# Parent-of-Origin inference for biobanks

Robin J. Hofmeister ®[1,2], Simone Rubinacci ®[1,2], Diogo M. Ribeiro ®[1,2], Alfonso Buil ®[3,4], Zoltán Kutalik ®[1,2,5] & Olivier Delaneau ®[1,2] ✉

Identical genetic variations can have different phenotypic effects depending on their parent of origin. Yet, studies focusing on parent-of-origin effects have been limited in terms of sample size due to the lack of parental genomes or known genealogies. We propose a probabilistic approach to infer the parent-of-origin of individual alleles that does not require parental genomes nor prior knowledge of genealogy. Our model uses Identity-By-Descent sharing with second- and third-degree relatives to assign alleles to parental groups and leverages chromosome X data in males to distinguish maternal from paternal groups. We combine this with robust haplotype inference and haploid imputation to infer the parent-of-origin for 26,393 UK Biobank individuals. We screen 99 phenotypes for parent-of-origin effects and replicate the discoveries of 6 GWAS studies, confirming signals on body mass index, type 2 diabetes, standing height and multiple blood biomarkers, including the known maternal effect at the *MEG3/DLK1* locus on platelet phenotypes. We also report a novel maternal effect at the *TERT* gene on telomere length, thereby providing new insights on the heritability of this phenotype. All our summary statistics are publicly available to help the community to better characterize the molecular mechanisms leading to parent-of-origin effects and their implications for human health.

Parent-of-Origin (PofO) effects refer to genetic variations having an effect on a phenotype that depends on the parent from which alleles are inherited[1,2]. PofO effects are thought to mainly result from genomic imprinting, a mechanism relying on parent-specific DNA methylation, named imprints, that silence one of the parental copies of a gene. Such parent-specific imprints are established during spermatogenesis and oogenesis and are maintained in all somatic cells of the offspring[3]. This leads to some genes, called imprinted genes, to exhibit an allele-specific expression pattern that depends on the PofO of the underlying genetic sequence. This allele-specific expression can be maintained throughout life or specific to some development states[4]. One of most studied imprinted loci in the human genome is probably the H19 loci at 11p15.5 that is involved in growth and development disorders such as the Beckwith–Wiedemann or Silver–Russel syndromes[5]. Multiple studies have investigated PofO effects on complex traits, notably for the

*KCNQ1* and *KLF14* genes whose associations with type 2 diabetes risk depends only on the maternal copies[6], as well as for the *MEG3/DLK1* imprinted locus associated with age at menarche[7] and platelet count[8].

Searching for PofO effects on a genome-wide scale requires knowing the PofO of each individual allele. The most direct approach to obtain this information relies on the availability of parental genomes, which allows using the Mendelian principles of inheritance to determine the parent from which a specific allele is inherited. Study cohorts usually include a small number of genotyped parent-offspring duos and trios, resulting in a low discovery power and a challenging detection of PofO effects. To alleviate this problem, multiple approaches have been explored so far. First, by deploying large efforts in data collection, such as the study performed on the DiscovEHR cohort[9], representing one of the largest PofO study done to date, with hundreds of phenotypes assessed for more than 22,000 samples with at

---

[1]Department of Computational Biology, University of Lausanne, Lausanne, Switzerland. [2]Swiss Institute of Bioinformatics (SIB), Lausanne, Switzerland. [3]Institute of Biological Psychiatry, Mental Health Services, Copenhagen University Hospital, Copenhagen, Denmark. [4]Lundbeck Foundation GeoGenetics Centre, GLOBE Institute, University of Copenhagen, Copenhagen, Denmark. [5]University Center for Primary Care and Public Health, University of Lausanne, Lausanne, Switzerland. ✉e-mail: olivier.delaneau@unil.ch

least one genotyped parent. Alternatively, this can also be achieved by meta-analysis across multiple cohorts regrouping duos and trios, with the caveat of restricting the analysis to the subset of phenotypes in common across datasets[7,10]. Second, statistical approaches have been proposed to test for PofO effects in large collections of unrelated samples by exploiting the differences in phenotypic variance between heterozygous and homozygous individuals, with the caveat of also detecting effects unrelated to PofO such as gene-environment interactions[11]. Third, it has been shown that the PofO of an individual's alleles can also be determined by the use of cousins as *surrogate parents* when parental genomes are not available[6]. This latter approach is particularly well suited for datasets comprising many samples from the same generation but also requires the genealogy of most individuals in the study cohort, which is not the case in large datasets such as the UK Biobank[12].

In this work, we present a probabilistic method to infer the PofO of alleles in biobank scale datasets from second- and third-degree relatives without requiring any parental genomes nor explicit genealogy to be known. To do so, our approach combines multiple estimation steps, involving surrogate parent groups formation, parental status assignment based on chromosome X, haplotype inference, Identity-By-Descent (IBD) detection, and haploid imputation. When applied to the UK Biobank dataset, this allows us to infer the PofO for 21,484 samples with high confidence in addition to the 4909 duos/trios for which we perform direct inference from parental genomes, resulting in a dataset comprising a total of 26,393 samples and 7.6 million variants. Considering duos/trios as the ground truth, we show that our PofO estimations from second- and third-degree relatives have a high call rate (~75%) and low error rate (<1%) at heterozygous genotypes. Taking advantage of the vast phenotypic diversity of the UK Biobank, we carry out genome-wide association scans for PofO effects for a total of 99 phenotypes, replicating well-known imprinted loci as well as discovering novel putative PofO associations, thereby demonstrating that our method has the potential to further reveal the contribution of PofO effects to complex traits. All the summary statistics for the conducted association scans are publicly available (http://poedb.dcsr.unil.ch/) and allow the exploration of the PofO effects for variants of interest across phenotypes.

## Results

### PofO inference from genotype data

To infer the PofO of all alleles carried by a given target sample, we proceed in two consecutive steps as detailed below:

1. *Identification of surrogate parents* (Fig. 1a). For each target sample (white British individual of the UK Biobank), we identify close relatives, and we determine which of the two parents (mother or father) conveys the relatedness. For this, we first look at pairwise kinship estimates given by KING[13] to identify second- or third-degree relatives and group them into the two parental groups based on their relatedness: they cluster in the same group if they are related and in different groups otherwise. Then, we assign parental status (maternal or paternal) to parental groups for male targets only by exploiting the fact that their single chromosome X copy is maternally inherited. Therefore, we search for relatives sharing portions of their chromosome X Identical-By-Descent (IBD) with the target sample and we label them as surrogate mothers. We also propagate the information to other relatives: those from the same parental group are also labeled as surrogate mothers and those from the other parental group as surrogate fathers. In case no IBD is found, we cannot annotate parental groups as maternal or paternal and we exclude the target sample from the dataset. Hereafter, we call *surrogate parents* the close relatives we identified using this approach.

2. *Assignment of PofO to alleles* (Fig. 1b). After the identification of surrogate parents, we assign PofO to the target's alleles. First, we

search for autosomal shared IBD segments between the target and the surrogate parents using IBD mapping robust to both phasing and genotyping errors (see **Methods**, Supplementary Fig. 1). Then, we classify the resulting IBD segments as being maternally or paternally inherited depending on the surrogate parent they map to. This delimits a subset of alleles that are co-inherited from the same parent within and across chromosomes (i.e., that co-localize on the transmitted set of homologous chromosomes). This leaves another subset of alleles for which we do not know the PofO (i.e., those not shared IBD with any of the surrogate parents). For those, we extrapolate the PofO using statistical phasing: we model alleles for which we know the PofO status as a haplotype scaffold[14] onto which all remaining alleles are probabilistically phased using SHAPEIT4[15] (Supplementary Fig. 1). The PofO assignment of these remaining alleles is then given by their frequency of co-localization onto each haplotype scaffold, which also reflects how reliable the phasing is (i.e., phasing certainty, Supplementary Fig. 1). Finally, we extrapolate the PofO for untyped variants by performing haploid imputation of each parental haplotype in turn using IMPUTE5[16] and the HRC as reference panel[17].

### Validation of the PofO inference on duos and trios

To assess the accuracy of our approach, we used 443,993 genotyped UK Biobank samples of British or Irish ancestry, together with their pairwise kinship estimates, to identify a subset of samples with parents and second-to-third degree relatives. For these samples, we inferred the PofO using two approaches: directly from the parents or using second-to-third degree relatives as surrogate parents. We compared the quality of the PofO inference given by surrogate parents to the direct approach based on parental genomes, considered to be the ground truth. We found a total of 3872 parent-offspring duos and 1037 trios, of which 1090 duos and 309 trios also have groups of surrogate parents. We used this subset of 1399 samples to assess optimal parameters and the accuracy of the method. We focused on two metrics: (i) the error rate, which is the percentage of heterozygous genotypes with incorrect PofO assignment and (ii) the call rate, which is the percentage of heterozygous genotypes for which a PofO call could be made (see **Methods**). We explored a range of different parameter settings for the IBD detection and PofO confidence score (i.e., phasing certainty onto the haplotype scaffold) and found that using haplotype segments longer than 3 cM as scaffold and a phasing certainty above 0.7 lead to a good trade-off between call rate and error rate (Fig. 2a). This resulted in a whole genome error rate of 0.51% and a call rate of 74.5%. As expected, the error and call rate depend on the number of available surrogate parents per target, with the call rate increasing and the error rate decreasing as the number of surrogate parents increases (Fig. 2b). The majority of our targets only have a single surrogate parent (75.95% of the target samples, Fig. 2c) and even in this case, a call rate of 70.9% and an error rate of 0.6% is achieved (Fig. 2b). We then considered the genomic localization of variants: we found a lower call rate and a slightly higher error rate as we approach telomeres, which results from phasing edge effects (Fig. 2d). Overall, we found small error rates for the majority of the variants: 79% have an error rate <1% and 56% inferred perfectly (Fig. 2e). This low error rate mostly results from the high phasing accuracy that can be achieved in the UK Biobank using SHAPEIT4[15]. Overall, we obtained a whole genome switch error rate of 0.0845% between consecutive heterozygous genotypes when comparing to parental genomes, with only small variations across chromosomes (Supplementary Fig. 2A). When looking at the distribution of these switch errors along the genome, we found that they mostly occur within small segments and that long range errors are almost entirely corrected by the use of haplotype scaffolds (Supplementary Fig. 2B). As a result, we obtained haplotypes that are resolved across entire chromosomes with only a few sporadic errors that, given their

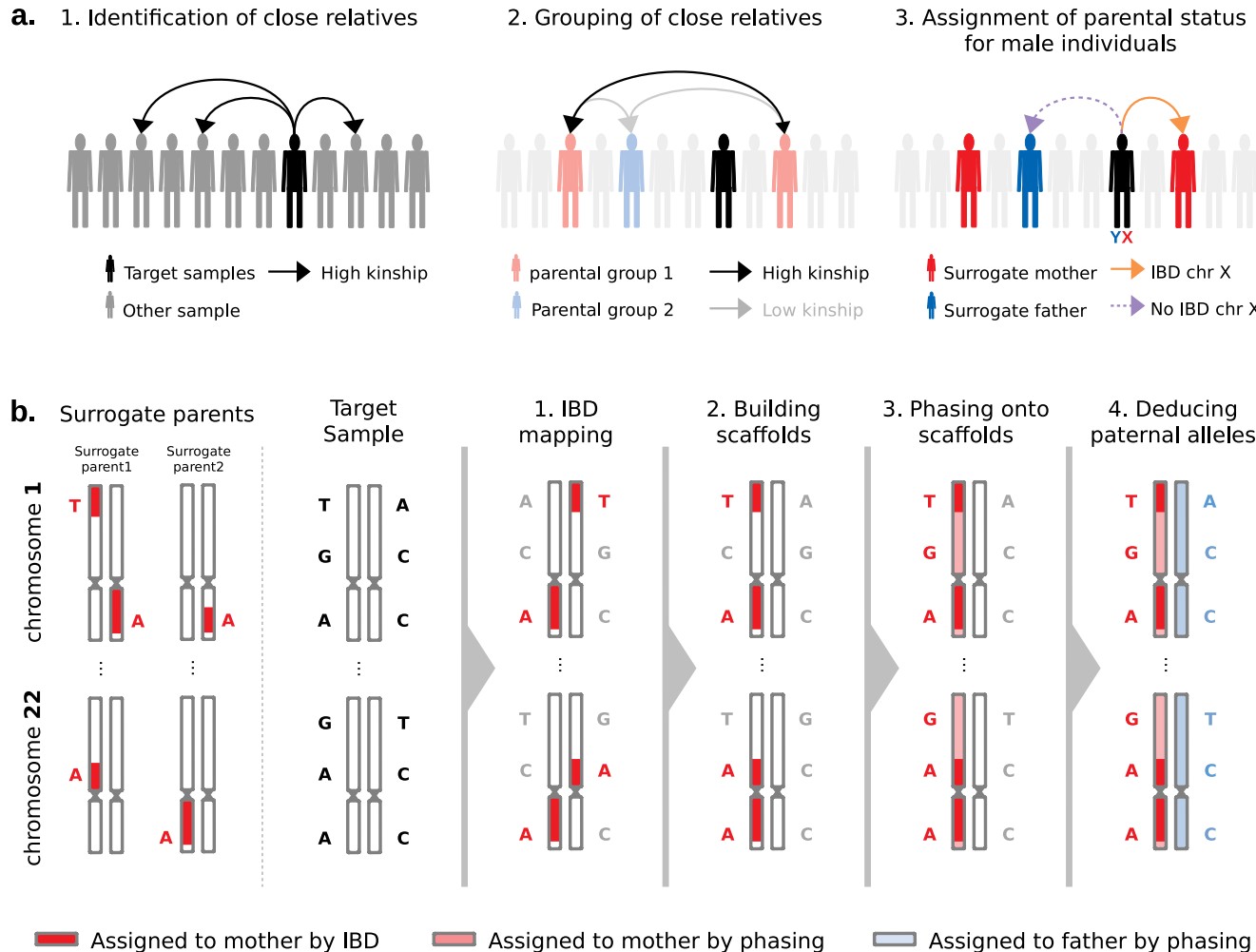

**Fig. 1 | Rationale of PofO inference. a** Identification of surrogate parents in 3 steps: (1) identification of close relatives for a target sample of interest using the pairwise kinship estimates, (2) clustering of close relatives by maximizing and minimizing the inter- and intra-groups relatedness, respectively, (3) assignment of parental status to close relatives' groups (i.e., surrogate parents) using IBD sharing on chromosome X for male targets. **b** Parent-of-origin inference in 4 steps: (1) identification of autosomal IBD segments shared between the target and the surrogate parents, (2) scaffold construction with co-inherited alleles localized on the same homologous chromosome across all autosomes, (3) statistical phasing of all remaining alleles against the scaffold and (4) whole genome deduction of the maternal and paternal origins of alleles from phasing probabilities.

frequency (<0.1% error rate), we believe to result mostly from genotyping errors.

## PofO inference in 26,393 individuals

For all genotyped British and Irish individuals in the UK Biobank without any genotyped parent ($N = 438,993$), we inferred the PofO using the method based on surrogate parents, as described above. In total, we found 105,826 samples with second-to-third degree relatives forming groups of surrogate parents. Amongst those, we could assign parental status to surrogate parent groups to a subset of 21,484 samples using IBD matching on chromosome X. Comparing the distribution of surrogate parents per target sample, we found a remarkable match between the full ($N = 21,484$) and the validation ($N = 1399$) datasets (Fig. 2c), suggesting that we can expect similar error rates between datasets. As our method requires IBD sharing between the targets and the surrogate parents, no inference can be made for chromosomes where no IBD sharing is found. The number of samples with PofO inference thus varies across chromosomes depending on their length, ranging from 15,645 samples (72.8%) for chromosome 21 to 20,381 samples (94.8%) for chromosome 1 (Fig. 2f). It follows that the call rate also varies across chromosomes, ranging from 66% for chromosome 21 to 77.9% for chromosome 2 (Fig. 2f). From the sample

point-of-view, we found that 31.3% of the samples have inference for the 22 autosomes and 96.1% have inference for more than 15 chromosomes (Supplementary Fig. 3). Finally, we merged the 21,484 samples with PofO inferred from surrogate parents together with the 4909 samples with PofO inferred from genotyped parents (3872 duos and 1037 trios) to get a final set comprising a total of 26,393 individuals with PofO inference (22,652 males and 3741 females) across 7.6 million variants genome-wide (Supplementary Table 1). Together with deep phenotyping provided by the UK Biobank, this represents a unique dataset to study PofO effects on complex traits.

## Discovery of PofO associations

Distinguishing paternally from maternally inherited alleles allows us to design different association scans to test the PofO specificity of associations: (i) maternal, to test only the maternally inherited alleles, (ii) paternal, to test only the paternally inherited alleles, (iii) differential, to compare paternally and maternally inherited alleles at heterozygous genotypes only and (iv) additive, as a control to test minor alleles regardless of PofO. Using these models we scanned for association 99 quantitative phenotypes of the UK biobank using BOLT-LMM[18] (Supplementary Data 1), for which we provide all summary statistics online (http://poedb.dcsr.unil.ch/).

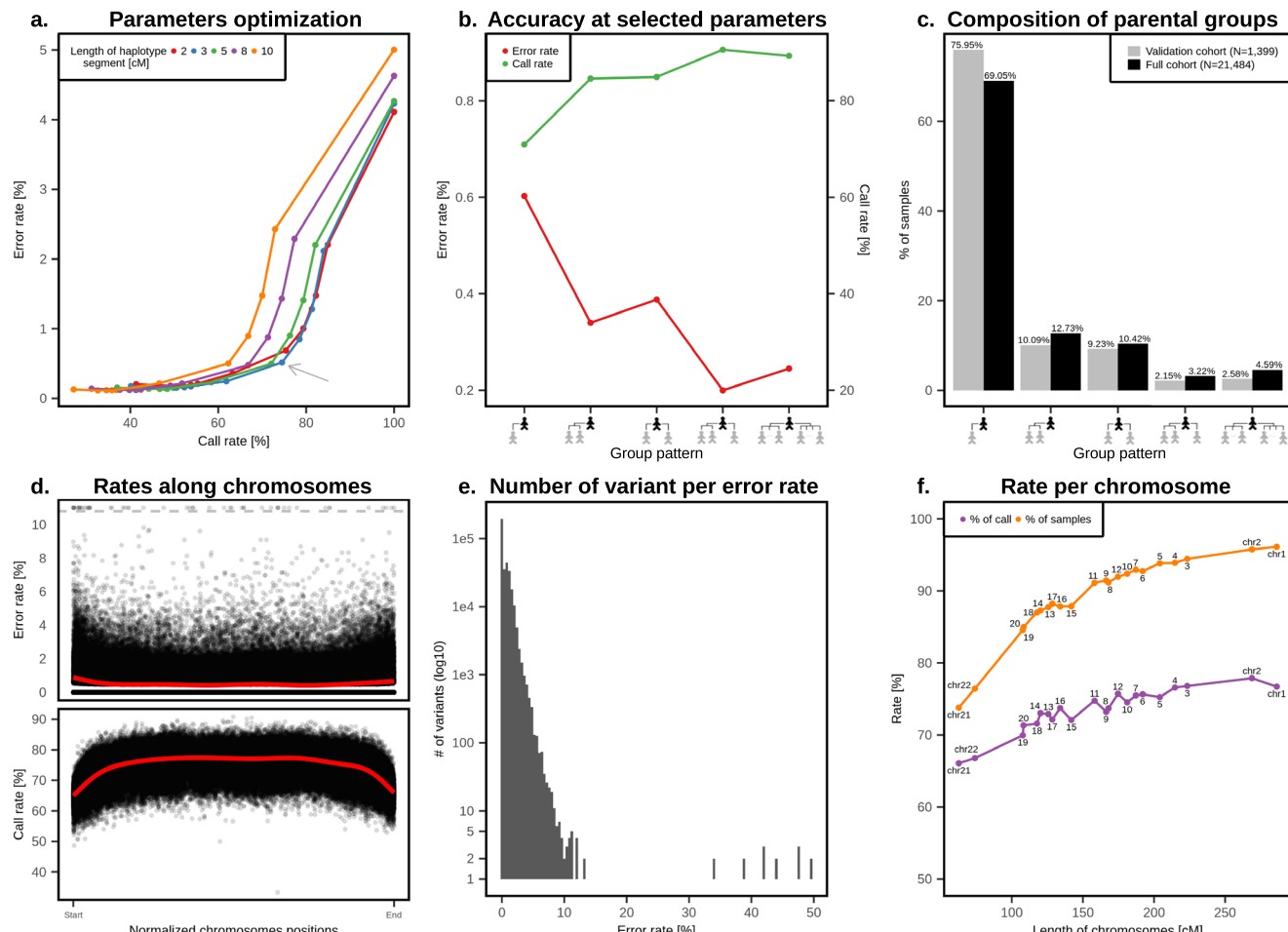

**Fig. 2 | Validation of the PofO inference. a** Call rate (*x*-axis) and error rate (*y*-axis) as a function of (i) the minimal length of IBD tracks for scaffold construction and (ii) the minimal phasing probability used to call a heterozygote as phased. Each point corresponds to a given phasing probability threshold going from 0.5 (right most point) to 1.0 (left most point) with steps of 0.05. The grey arrow indicates the parameters we used in our analysis (3 cM long IBD tracks and 0.7 minimal phasing probability). **b** Call rate (left *y*-axis) and error rate (right *y*-axis) as a function of the composition of the parental groups (*x*-axis). The latter ranges from one parental group with one surrogate parent (left) to two parental groups comprising multiple surrogate parents (right). **c** Fraction of targets as a function of the composition of the parental groups (*x*-axis): in the validation data (*N* = 1399) in gray and in the call set (*N* = 21,484) in black. **d** Error rate (top panel; *y*-axis) and call rate (bottom panel; *y*-axis) per variant site as a function of their normalized positions relative to each telomere (*x*-axes). Red lines are fitted density curves. Error rates greater than 10% are capped to 11% as indicated by the dashed gray line. **e** Distribution of error rates per number of variant sites (*y*-axis, log scale). **f** Fraction of samples (purple) and heterozygotes (i.e., call rate; orange) in the call set for which PofO is inferred, as a function of chromosome length (cM, *x*-axis). Chromosome numbers are shown next to the points in black. Source data are provided as a Source Data file.

In a first pass, we focused on variants being Bonferroni significant in both the differential and additive scans ($p < 5 \times 10^{-08}$) and used the paternal and maternal scans to determine the parental origin and the direction of the effect. We found two signals fulfilling these criteria. The first is a PofO association with platelet phenotypes at the *MEG3/DLK1* imprinted locus[19] (Table 1 and Fig. 3a, b). The lead SNP rs59228823 is an eQTL for *MEG3* in blood samples[20] and is associated with platelet count and platelet crit under the additive, maternal and differential scans but not under the paternal scan (Table 1). The minor allele C at this SNP significantly decreases the platelet count and crit when maternally inherited (Table 1 and Fig. 3c). A similar maternal effect has been previously reported on platelet count for another SNP in the same locus[8,9]: rs1555405, which is in linkage disequilibrium with rs59228823 ($R^2 = 0.59$). We also replicated the association at rs1555405 (maternal, paternal, differential *p*-values = $2 \times 10^{-16}$, 0.13, $1.4 \times 10^{-06}$, Supplementary Fig. 4A), suggesting that these two associations capture the same effect. The second PofO association we found is at SNP rs2735940 for the leukocyte Telomere Length (TL) phenotype, with the minor allele G decreasing TL only when maternally inherited (Table 1 and Fig. 4a–c). This SNP is located −1.5 kb upstream of the

promoter of *TERT* (Fig. 4b), a gene encoding for the catalytic subunit of the telomerase, an enzyme involved in TL maintenance[21]. This SNP is in high linkage disequilibrium with the SNP rs2853677 ($r^2 = 0.6$), previously reported in different GWAS for multiple cancers[22,23], blood cell counts[24], aging[25] and telomere length[26]. When directly testing rs2853677 in our data, we find a strong maternal effect similarly to the lead SNP (maternal, paternal, differential *p*-values = $4.6 \times 10^{-17}$, 0.8, $7.9 \times 10^{-11}$, Supplementary Fig. 4B). This suggests that a parent-of-origin effect underlies this pleiotropic locus.

In a second pass, we focused on associations that are Bonferroni significant in the differential scans but not supported by the additive scans. In total, we found 14 of these associations that we classified as putative PofO effects (Supplementary Table 2). This includes three maternal associations, four paternal associations, and seven associations with opposite effect between the paternally and maternally inherited alleles which are consistent with a pattern of bipolar dominance[2]. To confirm these results, we used a method developed by Hoggart et al.[11] designed to capture PofO effects by detecting an increased variance across heterozygous compared to homozygous. Using this method on the full set of British samples (*N* = 443,993), all

## Table 1 | Discovery of PofO associations

| Phenotypes | SNP | Chr | Position (hg19) | Risk allele | MAF | Add.P | Pat.P | Add.B | Pat.B | Mat.P | Mat.B | Diff.P | PofO effect | Mapped gene | UKB phenotype code | Hoggart et al. P. |
|---|---|---|---|---|---|---|---|---|---|---|---|---|---|---|---|---|
| Adjusted T/S ratio, Telomeres length | rs2735940 | 5 | 1296486 | G | 0.49 | 3.7e-08 | 0.46 | -0.049 | 0.008 | 2.1e-19 | -0.121 | 4.3e-13 | Maternal | TERT | 22191 | 0.00183 |
| Platelet crit | rs59228823 | 14 | 101185187 | C | 0.24 | 1.6e-10 | 0.66 | -0.065 | -0.007 | 6.6e-17 | -0.124 | 2.9e-09 | Maternal | MEG3, DLK1 | 30090 | 0.00180 |
| Platelet count | rs59228823 | 14 | 101185187 | C | 0.24 | 8.6e-10 | 0.58 | -0.064 | -0.011 | 6.8e-16 | -0.123 | 2.3e-08 | Maternal | | 30080 | 0.00180 |

PofO effects we identified based on a Bonferroni correction ($5 \times 10^{-08}$) on both the differential and the additive scans. Risk allele represents the minor allele tested in each of the four scans. Additive betas represent the phenotypic effects of minor alleles. Paternal betas represent the phenotypic effects of paternally inherited minor alleles. Maternal betas represent the phenotypic effects of maternally inherited minor alleles. Differential betas are not shown since they depend only on the parental alleles taken as reference. Genes were mapped using either eQTLs or ensembl Variant Effect Predictor (VEP). P-values are computed using BOLT-LMM[18], expect Hoggart et al.[1] p-values that are computed using the increased variance method (see Methods). Add Additive, Pat. Paternal, Mat. Maternal, Diff. Differential, P = p-values; B = betas; Chr=Chromosome.

associations have *p*-values <0.007 (Supplementary Table 2). The strongest opposite PofO effect involves the variant rs77403171 at 2q22.3, intronic to *ARHGAP15* and decreasing the eosinophil percentage when maternally inherited while increasing the trait when paternally inherited. *ARHGAP15* is a Rho GTPase-activating protein that has already been associated with multiple blood cell phenotypes, notably neutrophils, leukocytes, and eosinophils[27–29]. These are examples of genetic effects missed by the additive model as paternal and maternal contribution at heterozygous sites cancel out when considered together.

Finally, we used the PofO associations at the *MEG3/DLK1* locus and at the *TERT* locus to illustrate the benefit of using our PofO inference on the discovery power of PofO effects. To do so, we used 4909 UK Biobank duos/trios as baseline and gradually added random subsets of 5000 samples for which PofO inference could be made from surrogate parents, ending up with the full set of 26,393 samples. Doing so led to a clear boost in association strength for the additive, maternal and differential signals, with maternal scans ranging from non-significant on the duos/trios for both platelet crit and TL ($n = 4909$; *p*-value = $6.28 \times 10^{-04}$ and $8.36 \times 10^{-05}$, respectively) to strongly significant on the full sample size ($n = 26,393$; *p*-value = $6.6 \times 10^{-17}$ and $2.1 \times 10^{-19}$, respectively; Fig. 5a, b), while the paternal signal remained non-significant. Similarly, we also looked at the effects of errors in the PofO inference on the discovery power by randomizing the PofO assignment for an increasing number of samples. This progressively diluted the maternal signal onto the two parental origins while leaving the additive signal unchanged (Fig. 5c, d). Interestingly, the association with TL remains significant with up to 10% of errors in the PofO inference, suggesting that PofO testing could tolerate relatively high error rates with our sample size.

### Replication of PofO associations

The PofO callset for the UK Biobank generated with our method provides a powerful resource to replicate independent PofO associations or to annotate other types of associations as PofO effects. To show this, we assessed the ability of our method to replicate the results of seven GWAS studies across multiple phenotypes often studied in the context of PofO effects. These studies belong to three different categories: (i) PofO studies using trios or known genealogies, (ii) PofO studies across unrelated individuals using an increased-variance method, and (iii) studies investigating genotype-environment (GxE) interactions.

**Standing height.** We focused on the 11 PofO associations reported in three studies making use of genealogy-based PofO inference[30–32], 9 of which could be assessed in our data (identical SNP-phenotype pair in the UK Biobank). Seven of these associations are located in two well-known imprinted regions, 11p15.5 and 14q32, and the remaining two are located in the HLA region. Only one association has been replicated across the two of the three studies at rs143840904. In contrast, we replicated 8 associations out of the 9 we could test, with the same parent and direction of effects (Table 2A–C), thereby reinforcing the role of these two well-known imprinted regions on height and providing further evidence on the PofO effect at the HLA region.

**Blood biomarkers.** A recent study[9] examined multiple blood biomarkers and reported a total of 10 PofO associations within imprinted loci using trios-based PofO inference. In our dataset we were able to assess 7 of these associations and replicated 5 of them, with the same parent and direction of effects (Table 2D). This included the PofO effect on platelet phenotypes at the *MEG3/DLK1* locus we reported earlier.

**Type 2 diabetes.** Kong et al.[6] reported a total of 4 PofO associations on type 2 diabetes (T2D) using genealogy-based PofO inference. They fall within two distinct regions that harbor well-documented imprinted

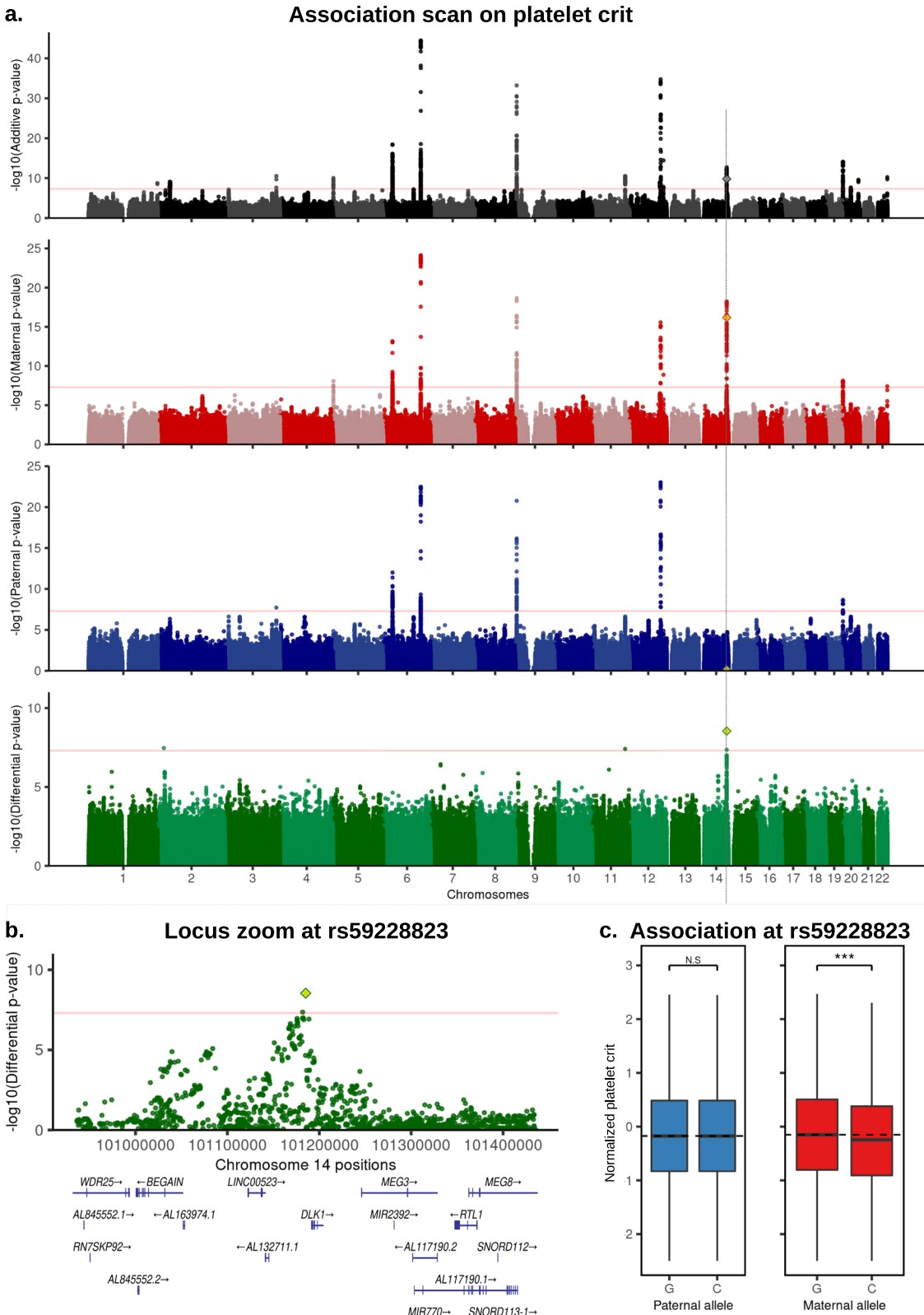

**Fig. 3 | Association scans for PofO effects on platelet crit. a** Manhattan plots of four association scans with platelet crit. From top to bottom plots are shown results for additive (black), maternal (red), paternal (blue) and differential (green) scans. The lead variant mentioned in this study (rs59228823) is shown with a diamond. Red horizontal lines indicate genome-wide significance threshold at $-\log10(5 \times 10^{-08})$. **b** Locus zoom at rs59228823 on the differential scan. **c** Box plot of the normalized platelet crit (*y*-axis) stratified by risk alleles and origin at SNP rs59228823; paternal in blue and maternal in red (*x*-axis). The horizontal dotted lines represent the phenotypic median of the major allele G. Boxes bound the 25th, 50th (median), and the 75th quantiles. Whiskers range from minima (lower) to maxima (upper). Sample sizes are $n_{paternal}$(G/C) = 16,285/4,769 and $n_{maternal}$(G/C) = 16,368/4686 individuals. N.S non-significant (*p*-value = 0.66); ***=significant (*p*-value = $6.6 \times 10^{-17}$) (computed with BOLT-LMM[18]). Source data for (**a**) and (**b**) are provided as a Source Data file.

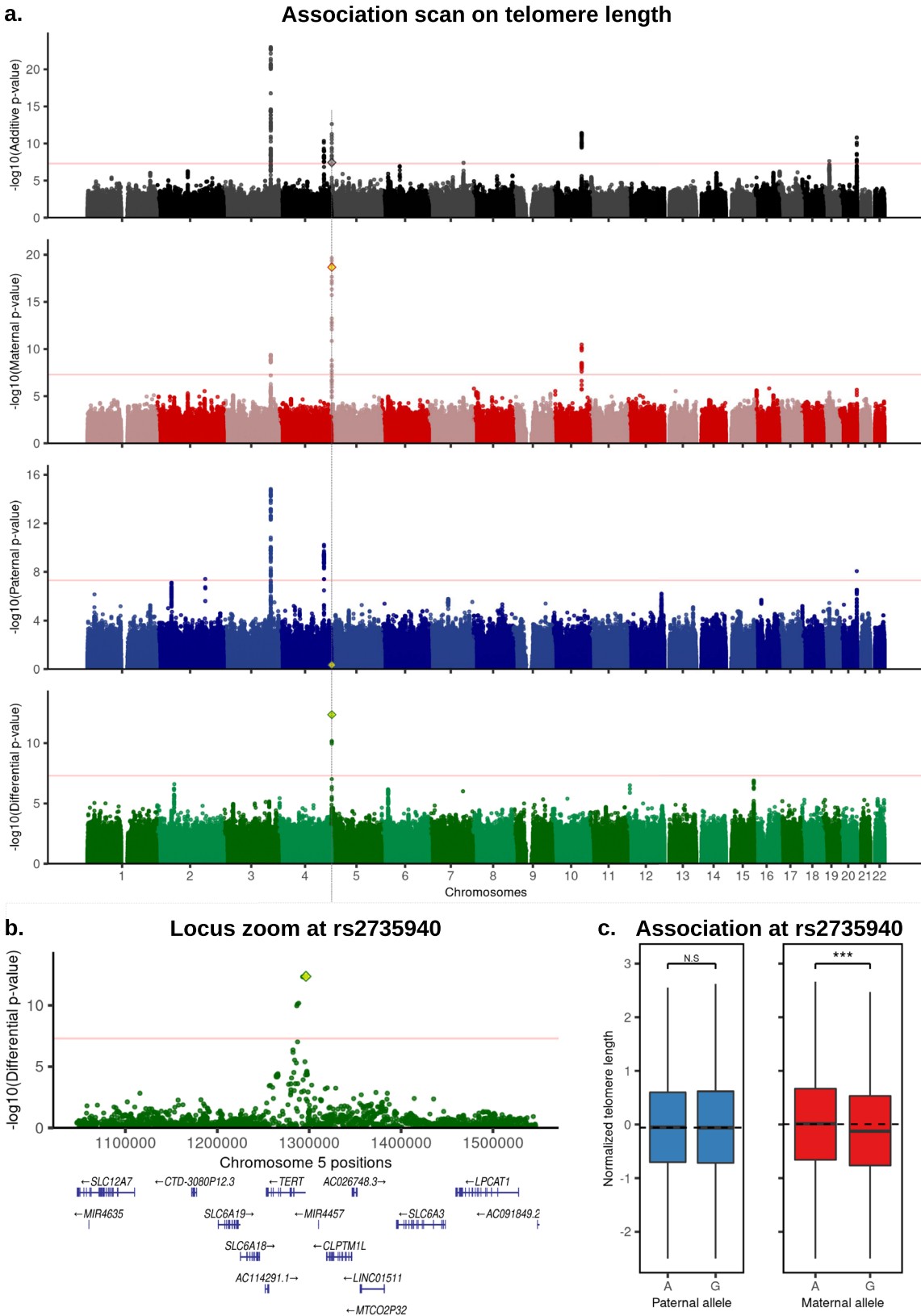

**Fig. 4 | Association scans for PofO effects on telomere length. a** Manhattan plots of four association scans with telomere length. From top to bottom plots are shown results for additive (black), maternal (red), paternal (blue) and differential (green) models. The lead variant mentioned in this study (rs2735940) is shown with a diamond. Red horizontal lines indicate genome-wide significance threshold at −log10($5 \times 10^{-08}$). **b** Locus zoom at rs2735940 on the differential scan. **c** Box plot of the normalized telomere length (*y*-axis) stratified by risk alleles and origin at SNP rs2735940; paternal in blue and maternal in red (*x*-axis). The horizontal dotted lines represent the phenotypic median of the major allele A. Boxes bound the 25th, 50th (median), and the 75th quantile. Whiskers range from minima (lower) to maxima (upper). Sample sizes are $n_{paternal}$(A/G) = 10,627/10,337 and $n_{maternal}$(A/G) = 10,635/10,329 individuals. N.S non-significant (*p*-value = 0.46); *** = significant (*p*-value = $2.1 \times 10^{-19}$) (computed with BOLT-LMM[18]). Source data for (**a**) and (**b**) are provided as a Source Data file.

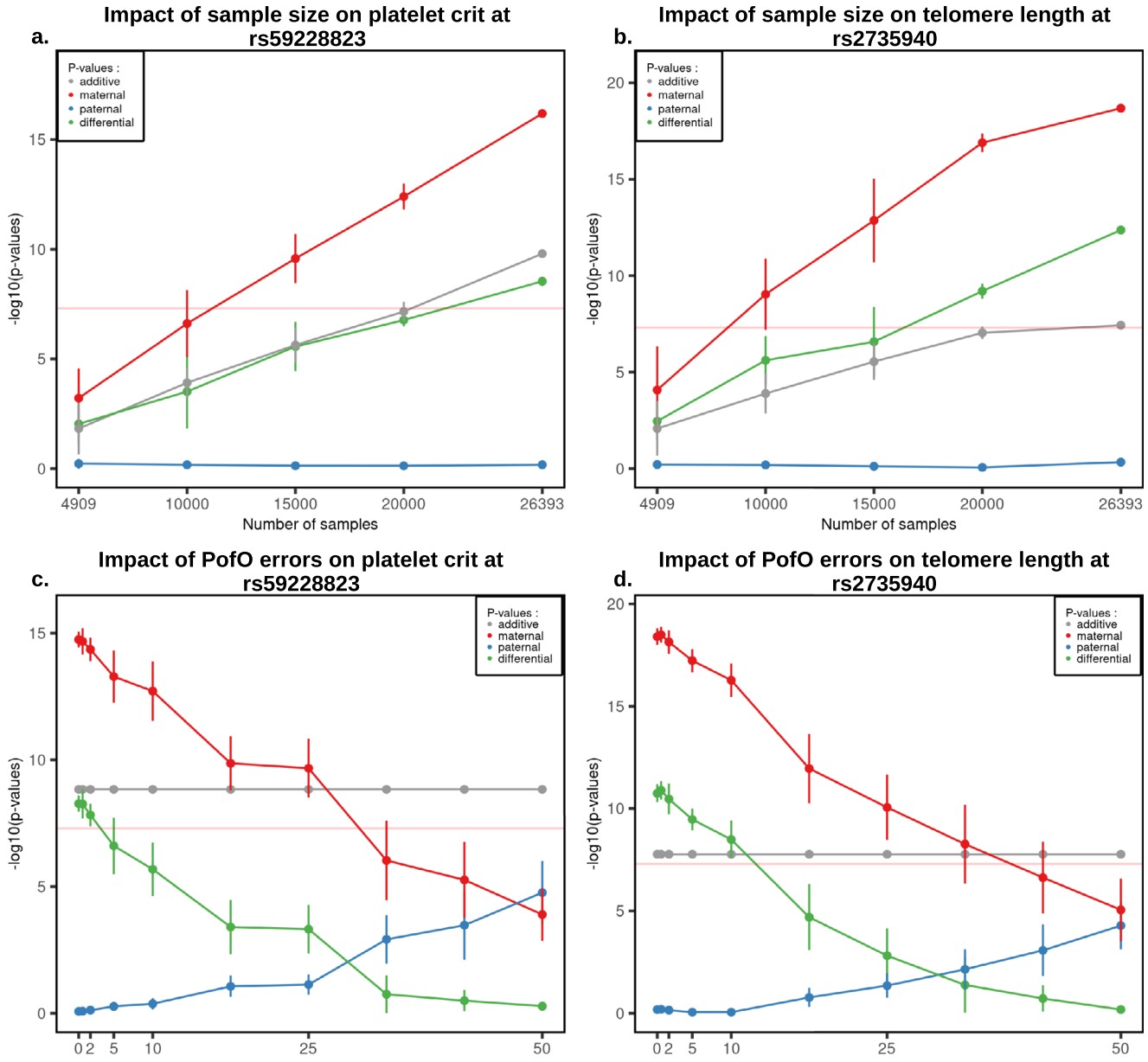

**Fig. 5 | Robustness of the PofO testing. a**, **b** Association strength as −log10(*p*-value) for rs59228823 and rs2735940 (*y*-axis) on platelet crit and TL, respectively, as a function of the number of randomly chosen samples included in the analysis under the additive (black), paternal (blue), maternal (red) and differential (green) scans. Each point for *N* = [10,000; 15,000; 20,000] represents the median *p*-value obtained after 10 randomizations with vertical bars representing the standard error. Points for *N* = 4909 and *N* = 26,393 represent the *p*-values obtained using only the samples with genotyped parents and using our full sample size,

respectively. **c**, **d** Association strength as −log10(*p*-value) for rs59228823 and rs2735940 (*y*-axis) on platelet crit and TL, respectively, as a function of the fraction of samples for which PofO has been randomly drawn (*x*-axis, 100% = 26,393). Samples included are those for which the PofO has been inferred from the surrogate parents. Each point represents the median *p*-value obtained after 10 randomizations with vertical bars representing the standard errors. *P*-values are computed with BOLT-LMM[18]. Source data are provided as a Source Data file.

gene clusters, 11p15.5[33] and 7q32[34,35]. As we could not directly test T2D status due to the small number of cases in our dataset, we tested the biomarker most correlated with T2D: glycated hemoglobin (HbA1c, https://ukbb-rg.hail.is/). By doing so, we replicated the three strongest associations with the same parental effect (Table 2E). In addition, we phenome-wide analyzed these four variants in our dataset and found 22 associations with differential *p*-value <0.01 (Supplementary Data 2) for 20 distinct phenotypes, many of them closely related to T2D. This illustrates how the deep phenotyping of UK Biobank can help to provide new mechanistic insights for these four T2D risk alleles at the biomarker level.

**BMI by increased variance.** Hoggart et al.[11] reported a total of 6 PofO associations with BMI using an increased-variance method designed to capture PofO effects, two of which were replicated using independent family datasets. These include variants associated with known imprinted genes, *SLC2A10* at 20q13.12 and *KCNK9* at 8q24.3. We could replicate the strongest association in our dataset at the *KCNK9* locus, with the T allele of rs2471083 increasing BMI when maternally inherited (Table 2F). Here, our replication offers additional support for the KCNK9 locus and confirmation of the maternal origin of this effect, an information that the increased-variance approach can not provide.

## Table 2 | Replication of PofO associations

| Study | SNP | External studies | | | | | | | Our study | | | | | | | | Locus |
|---|---|---|---|---|---|---|---|---|---|---|---|---|---|---|---|---|---|
| | | Add.P | Add.B | Pat.P | Pat.B | Mat.P | Mat.B | Diff.P | UKB phenotype code | Add.P | Add.B | Pat.P | Pat.B | Mat.P | Mat.B | Diff.P | |
| A. Benonisdottir et al.[31] | **rs147239461** | 2.8e-03 | **-0.043** | **5.9e-13** | -0.12 | 9.4e-04 | 0.056 | 1.2e-13 | Standing height: 50 | 0.5 | -0.0095 | **0.017** | **-0.05** | 0.076 | 0.044 | 3.5e-03 | IGF2, H19 |
| | rs7482510 | 4.5e-04 | -0.03 | 5.1e-11 | -0.065 | 0.076 | 0.018 | 4.7e-09 | | 0.011 | -0.019 | 0.055 | -0.02 | 0.18 | -0.016 | 0.7 | |
| | **rs143840904** | 1.3e-05 | -0.11 | 0.042 | 0.057 | **2.0e-17** | **-0.26** | 1.6e-14 | | 2.2e-06 | -0.1 | 0.03 | -0.06 | **6.2e-07** | **-0.16** | 0.041 | KCNQ1 |
| | **rs41286560** | 0.078 | **-0.031** | **2.2e-08** | -0.12 | 1.7e-03 | 0.067 | 7.4e-10 | | 0.47 | -0.013 | **0.02** | **-0.06** | 0.32 | 0.025 | 0.018 | RTL1 |
| B. Zoledziewska et al.[30] | **rs143840904** | 4.58e-05 | -0.152 | 0.9653 | -0.0021 | **3.92e-08** | **-0.274** | 7.55e-05 | Standing height: 50 | 2.2e-06 | -0.10 | 0.03 | -0.069 | **6.2e-07** | **-0.16** | 0.041 | KCNQ1 |
| | **rs2075870** | 2.65e-05 | -0.158 | 0.793 | -0.0172 | **6.97e-08** | **-0.273** | 2.0e-04 | | 1.3e-04 | -0.07 | 0.1 | -0.05 | **9.6e-07** | **-0.15** | 0.03 | |
| | **rs149658560** | 1.01e-04 | -0.161 | 0.8183 | -0.0121 | **2.93e-07** | **-0.297** | 3.0e-04 | | 0.11 | -0.02 | 0.85 | 0.009 | **1.0e-04** | **-0.108** | 4.6e-03 | |
| | **rs67004488** | 1.2e-06 | -0.157 | 0.3875 | -0.40 | **5.21e-07** | **-0.244** | 2.4e-03 | | 7.2e-04 | -0.04 | 0.024 | -0.055 | **4.7e-04** | **-0.074** | 0.33 | |
| C. Granot-Hershkovitz et al.[32] | **rs1042136** | 1.82e-02 | -0.006 | **1.55e-08** | -0.023 | 4.21e-01 | 0.005 | 1.39e-04 | Standing height: 50 | 0.3 | -0.0072 | **0.036** | **-0.025** | 0.47 | 0.010 | 0.064 | HLA |
| | **rs1431403** | 8.54e-05 | -0.004 | **5.41e-06** | -0.011 | 5.88e-03 | -0.007 | 6.72e-03 | | 0.097 | -0.010 | **0.014** | **-0.023** | 0.85 | 0.003 | 0.052 | |
| | rs9332053 | 4.85e-01 | 0.153 | 2.28e-01 | 1.286 | 9.59e-06 | -4.241 | 3.17e-03 | Hip circumference: 49 | 0.38 | 0.025 | 0.35 | 0.039 | 0.87 | -0.001 | 0.45 | RCBTB2 |
| D. Kim et al.[9] | rs117989553 | 9.6e-27 | -0.09 | 1.9e-05 | -0.13 | 0.15 | -0.04 | - | Monocyte percentage: 30190 | 3.9e-10 | -0.113 | 1.6e-04 | -0.096 | 5.6e-06 | -0.126 | 0.59 | GRB10 |
| | **rs11742106** | 3.2e-19 | -0.12 | **7.9e-05** | -0.19 | 0.084 | -0.08 | - | | 4.7e-04 | -0.089 | **1.7e-04** | -0.138 | 0.27 | -0.05 | 0.087 | |
| | rs117515500 | 1.1e-07 | 0.07 | 0.14 | 0.07 | 1.5e-05 | 0.2 | - | Cholesterol: 30690 | 0.66 | -0.011 | 0.74 | -0.009 | 0.67 | -0.026 | 0.82 | IGF2R |
| | **rs12154627** | 1.2e-23 | 0.04 | 0.064 | 0.03 | **2.1e-16** | **0.13** | - | HDL-C: 30760 | 0.0018 | 0.031 | 0.6 | 0.001 | **2.7e-07** | **0.066** | 3.4e-04 | CPA4, MEST, KLF14 |
| | **rs12154627** | 5.1e-17 | -0.04 | 0.56 | -0.01 | **1.4e-09** | **-0.1** | - | Triglycerides: 30870 | 5.7e-04 | -0.035 | 0.42 | -0.015 | **1.6e-05** | **-0.06** | 0.055 | |
| | **rs4758459** | 8.2e-09 | 0.03 | 0.073 | 0.03 | **5.3e-05** | **0.07** | - | Total bilirubin: 30840 | 0.061 | 0.014 | 0.89 | 0.001 | **3.5e-04** | **0.044** | 0.0097 | H19, IGF2 |
| | **rs10146962** | 3.2e-20 | -0.04 | 0.12 | 0.02 | **2.3e-11** | **-0.1** | - | Platelet crit: 30090 | 1.5e-11 | -0.062 | 0.14 | -0.021 | **7.0e-17** | **-0.112** | 6.5e-07 | DLK1, MEG3 |
| E. Kong et al.[6] | **rs2237892** | 0.043 | 1.15 (OR) | 0.24 | 1.03 (OR) | **0.0084** | 1.30 (OR) | 0.054 | Glycated hemoglobin (HbA1c): 30750 | 0.34 | -0.021 | 0.14 | 0.037 | **0.047** | -0.061 | 0.014 | KCNQ1 |
| | rs231362 | 0.013 | 1.10 (OR) | 0.73 | 0.98 (OR) | **6.2e-05** | 1.23 (OR) | 0.0032 | | 0.0044 | -0.029 | 0.45 | -0.015 | **4.7e-04** | -0.050 | 0.039 | KCNQ1 |
| | **rs2334499** | 0.034 | 1.08 (OR) | **4.7e-10** | 1.35 (OR) | 0.002 | 0.86 (OR) | 4.1e-11 | | 0.13 | 0.011 | **4.3e-04** | 0.043 | 0.17 | -0.0184 | 2.6e-04 | HCCA2 |
| | rs4731702 | 0.039 | 1.08 (OR) | 0.79 | 0.99 (OR) | 0.001 | 1.17 (OR) | 0.022 | | 0.011 | -0.023 | 0.061 | -0.021 | 0.063 | -0.028 | 0.99 | KLF14 |
| F. Hoggart et al.[11] | **rs2471083** | **9.34E-07 (p-value PofO effect)** | | | | | | | Body mass index (BMI): 21001 | 0.039 | 0.019 | 0.45 | -0.009 | **1.2e-04** | 0.055 | 5.5e-04 | KCNK9 |
| | rs3091869 | 4.7E-06 (p-value PofO effect) | | | | | | | | 0.81 | -0.002 | 0.76 | -0.003 | 0.92 | -0.0009 | 0.84 | SLC2A10 |
| G. Kerin et al.[36] | **rs539515** | **p-value GxE = 6.5e-12 beta GxE = -0.0117** | | | | | | | Body mass index (BMI): 21001 | **1.3e-07** | 0.052 | **7.3e-09** | 0.083 | 0.047 | 0.028 | 0.0069 | SEC16B |
| | rs2153960 | p-value GxE = 6.5e-9 beta GxE = -0.0098 | | | | | | | | 0.31 | -0.009 | 0.33 | -0.014 | 0.62 | -0.006 | 0.72 | FOXO3 |

For trios-based and genealogy-based PofO associations (A–E), we considered as replicated associations for which the same parental effect and direction of the effect could be retrieved. For the increased variance method (F) and the GxE interactions (G), we considered associations as being PofO effect when only one parental scan was significant ($p < 0.05$). Additive betas represent the phenotypic effects of minor alleles. Paternal betas represent the phenotypic effects of paternally inherited minor alleles. Maternal betas represent the phenotypic effects of maternally inherited minor alleles. Differential betas are not shown since they depend only on the parental alleles taken as reference. Genes were mapped by the respective studies. P-values (our study) are computed using BOLT-LMM[18]. *Add* Additive, *Pat* Paternal, *Mat* Maternal, *Diff.* Differential, *P* = p-values; *B* = betas. Bold indicate replicated associations.

**BMI by GxE.** We hypothesized that some GxE signals detected for BMI could be due to PofO effects. Using the UK Biobank, Kerin et al.[36] reported two GxE associations at rs2153960 and rs539515, mapping to *FOXO3* and *SEC16B*, respectively, with the latter replicated by another study[37]. In our study, we found a paternal effect of rs539515 on BMI (Table 2G, maternal, paternal, differential *p*-values = 0.047, $7.3 \times 10^{-09}$, 0.0069). Interestingly, when performing a phenome-wide scan of the 1q25.2 locus harboring rs539515, we found paternal associations between four SNPs in high LD (rs527065, rs539515, rs8030 and rs531385; $r^2 > 0.5$) with weight, waist circumference, hip circumference, basal metabolic rate and arm/leg mass (maternal *p*-values > 0.05, paternal *p*-values $<5 \times 10^{-8}$, differential *p*-values <0.005; Supplementary Table 3). All these SNPs also map to *SEC16B* (either intronic or splicing QTLs) and have already been associated with weight- or obesity-related phenotypes under the additive model[38–40]. Altogether, this suggests that the GxE effect of *SEC16B* on BMI is likely due to a paternal effect.

**Birth weight.** We investigated the 22 PofO associations reported with birth weight phenotype[41] and were not able to replicate any of these associations in our data, nor any of the additive ones (Supplementary Data 3). This is most likely due to some phenotype misspecification of birth weight in the UK Biobank as this phenotype is self-reported by individuals between 39 and 73 years old, which is likely less reliable than newborn birth weight reported by the mother. Additionally, the individuals with available birth weight specification represent only half of our samples (Supplementary Data 1) which considerably decreases the discovery power.

## Discussion

Studying PofO effects requires parental genomes or genealogies to determine the set of alleles transmitted to the offspring by each of the two parents. As a consequence, this prevents the study of PofO effects in biobanks, usually comprising a large and diverse panel of phenotypes. In this work, we propose an approach that leverages the high degree of relatedness between individuals inherent to biobank-scale datasets in order to infer the PofO of alleles for many individuals and variant sites without any parental genomes or genealogy being available. When applied on the UK Biobank, this approach could predict the PofO of alleles for around 5% of the total number of samples, resulting in a dataset comprising the PofO inference for more than 26,000 samples at 7.6 million variants. Together with deep phenotyping, this dataset allows studying PofO effects on a large scale with improved discovery power, as demonstrated by our ability to replicate many known PofO associations as well as discover new ones.

We looked at PofO associations at three different levels. First, we reported two clear PofO associations supported by additive signals: a maternal effect on platelet phenotypes located in the *MEG3/DLK1* imprinted locus that has already been described[8,9] and another maternal effect on TL at the *TERT* locus, a gene repeatedly associated with TL under an additive model. This new PofO signal at the *TERT* locus is particularly interesting, not only for its implication in cancer[42], but also because TL has been found to be highly heritable and proposed to be under imprinting mechanisms[43–47], which has not yet been confirmed. In this work, we highlight a strong maternal genetic effect at the *TERT* locus, thereby providing additional evidence of the parent-of-origin component in TL heritability and hypothesis on the imprinting status of *TERT*. In addition to this, we also reported 14 new putative PofO associations across multiple complex traits and confirmed them by replicating the signals in a larger UK Biobank sample set using an increased variance method. These new associations represent interesting candidates of PofO effects in the human genome and would deserve further investigation and replication in independent datasets. Interestingly, none of

them fall in imprinting regions, suggesting that the current annotation of imprinted genes is still incomplete or that the molecular mechanisms underlying PofO effects are not necessarily directly linked to genomic imprinting[48]. Finally, we replicated the results of 6 GWAS on PofO out of the 7 we investigated, confirming PofO effects on BMI, T2D, standing height and multiple blood biomarkers. We also showed that the summary statistics we provide can be used to annotate additive signals (e.g., *TERT*) or variance QTL (GxE, e.g., *SEC16B*) as PofO. We believe that an increase of power is still necessary to detect additional PofO effects with strong confidence but that the current approach already provides a useful resource that can reveal many other associations by meta-analysis. Besides, we also believe that our dataset can be used for more targeted GWAS scans and reveal new putative PofO effects by focusing only on known imprinted loci, only on additive associations or on both criteria together[6,9], thereby decreasing the cost of multiple testing corrections.

One of the strengths of our PofO inference method resides in its ability to make PofO calls with a low error rate. Regardless, the presence of errors in the inference is unlikely to produce false positive PofO associations, but only decrease the statistical power of the study, since inference errors are expected to be drawn independently from the phenotypes. Instead, errors are expected to lead to false negatives as PofO signals get diluted onto the two parental origins and thus decrease association power. In this work, we controlled for this by focusing exclusively on high-confidence PofO calls, which corresponds to a call for 74.5% of heterozygous genotypes with an estimated error rate below 1%. The overall high accuracy in our estimates could be achieved thanks to recent progress in the statistical estimation of haplotypes for very large sample sizes[15,49] so that the PofO status inferred within IBD tracks could be confidently propagated to entire chromosomes. Further improvements in phasing algorithms could be made by explicitly modeling IBD sharing between close relatives, eventually through inter- and intra-chromosomal scaffolding as we performed in this work.

Our ability to infer PofO depends on the availability of close relatives. Surprisingly, even when only a single third-degree relative is available for IBD mapping, we achieve a high call rate and a low error rate. We believe this could be further improved by using more distant relatives, even if they will contribute less to the inference than second- and third-degree relatives. In addition, our PofO inference depends on the ability to assign parental status to relatives based on IBD sharing on chromosome X, which comes with some flaws. First, our current inference is only possible for males as it leverages chromosome X haploidy, which means that only non-sex specific and male specific PofO effects can be investigated. As a results, female specific PofO effects, which could be of great interest given the recent findings on sexual dimorphisms[50], notably for anthropomorphic traits, are likely missed by this approach. Potential improvements should come with whole genome sequencing (WGS) data: parental status assignment based on rare variant matching on chromosome Y and mitochondrial DNA would likely become possible. In the UK Biobank, this has the potential to substantially increase the sample size above the ~26,000 samples we have so far to a theoretical upper bound of 105,826 samples, which corresponds to the number of samples for which we found groups of close relatives in the dataset. This could further boost the discovery power of downstream PofO association scans. Second, this approach can be confounded by high levels of inbreeding which could lead a sample to share portions of the chromosome X IBD with close relatives on both sides of the family, therefore greatly complexifying sex assignment. However, we consider this issue to be almost negligible in this study as the UK biobank mostly comprises outbred individuals. Conversely, admixture affects kinship estimation and therefore our ability to find surrogate parents, although this can be compensated

by using a robust method for kinship estimation in admixed populations[51].

Overall, this study is a valuable resource to further characterize PofO effects and investigate the impact of imprinting genes on complex traits. Although the multiple successive steps of this approach (IBD mapping, phasing, imputation) are difficult to fully automatize, we expect it to be applicable to other biobanks, such as those collected by the FinnGen research project (https://finngen.gitbook.io/documentation/), the Million Veteran Program[52] or The Estonian Biobank[53]. Collective efforts would allow the detection of PofO effects with an unprecedented sample size by meta-analyzing PofO effects across multiple biobanks and therefore greatly help future research on the molecular mechanisms leading to PofO effects and their implication for human health.

## Methods

### Duos/Trios identification
To identify trios and duos we used pairwise kinship and IBS0 estimates up to third degree relative computed using KING[13] and provided as part of the UK biobank study. Following Manichaikul et al.[13] and Bycroft et al.[12], we defined offspring-parent pairs as having a kinship coefficient between 0.1767 and 0.3535 and an IBS0 below 0.0012 (Supplementary Fig. 5). We also added the condition of age difference greater than 15 years between parent-offspring pairs. We used the age and sex of the individuals to distinguish parents and offspring. For the trios, we also ensured that the two parents have different sex. Starting from 147,731 UKB individuals with at least one third degree relative, we found a total of (i) 1064 samples with both mother and father (i.e., trios) and (ii) 4123 samples with mother or father (i.e., duos). We used the reported ancestry of individuals to keep only genotyped individuals of British and Irish ancestry ($N = 443,993$), which resulted in 1037 trios and 3872 duos.

### IBD based group inference
We used pairwise kinship and IBS0 estimates up to the third degree relative to identify sibling pairs (kinship between 0.1767 and 0.3535 and IBS0 above 0.0012), and second- and third-degree relatives' pairs (kinship below 0.1767) for all genotyped individuals of British and Irish ancestry ($N = 443,993$) (Supplementary Fig. 5). For the following steps, we used only second- and third-degree relatives to form surrogate parent groups. We excluded siblings as they share the same two parental genomes and therefore are not informative to distinguish the paternal from the maternal genome. We found 106,414 individuals with at least one second or third degree relative and 21,255 sibling pairs. For individuals with two or more second- and third-degree relatives, we separated those relatives into groups, representing the groups of relatives on each side of the family (i.e., mother-side relatives and father-side relatives). To do so, we used the relatedness in-between these relatives: those related to each other are expected to be on the same side of the family, while those unrelated to each other are expected to be on different sides of the family. We built for each individual a kinship symmetric matrix of size $N \times N$, where $N$ is the number of second-to-third relatives of the target individual considered, filled with the kinship values in-between each relative. We then used the 'igraph' R package to cluster these relatives into groups based on their relatedness similarly to what has been done by Bycroft et al.[12]. As we wanted a maximum of two distinct groups (i.e., one paternal and one maternal), we excluded samples with more than two clusters of relatives from the analysis as it indicates ambiguous cases. Similarly, if a second-to-third degree relative is related to the two clusters, we also excluded the sample to avoid ambiguous cases. Importantly, this is often a symmetric assignment: when A is part of the group of relatives of B, this usually involves B is part of the relatives of A. We identified a total of 105,826 individuals with groups of relatives, ranging from one group of one relative to two groups of

more than two relatives. This includes 309 individuals having also both parents genotyped (i.e., trios) and 1090 having a single parent genotyped in the data (i.e., duos). These 1399 individuals with at least one genotyped parent and groups of close relatives constitute our validation data set on which we applied our PofO inference method using the close relatives as surrogate parents, ignoring the parental genomes. We then used parental genomes to compute the accuracy of our inference.

### Group assignment
We assigned parental status (i.e., mother or father) to groups of close relatives by examining shared IBD segments on chromosome X using XIBD[54], a software specifically designed to map IBD on chromosome X (Fig. 1c). This assignment was only possible for males as they inherit their only chromosome X copy from their mother: a close relative, male or female, sharing IBD on chromosome X with the target is expected to be from the maternal side of the family. To empirically determine the IBD threshold above which only mother-side relatives are found, we used the 1399 samples of our validation set (i.e., with close relatives' groups and genotyped parents). We computed the IBD sharing on chromosome X for each target-relative pair, knowing the correct parental side of the relatives from the kinship in between the relatives and the available parents. We found that only mother-side relatives share more than 0.1 of IBD1 on chromosome X (Supplementary Fig. 6), a value that we used as a threshold to assign maternal status. Across the 107,038 individuals having groups of close relatives, 48,814 individuals are males, and we assigned the group of close relatives to the maternal side of the family for 20,620 of them. By extension, we propagated the maternal status to the relatives from the same parental group, and we labeled as paternal the relatives from the other group. We then used the underlying idea that siblings share the same set of cousins, uncle, and aunt to enrich our set of samples. We searched for siblings of these 20,620 individuals having the exact same close relatives' groups. We found 864 such siblings, resulting in a total of 21,484 individuals with close relatives' groups assigned to parental status (i.e., surrogate parents). Notably, this strategy allowed us to assign parental status for a small additional subset of female individuals ($N = 775$, Supplementary Table 1).

### Genotype processing
We used the UK biobank SNP array data in PLINK format. We converted the UK biobank PLINK files into VCF format using PLINK v1.90b5[55], which resulted in 784,256 variant sites across the autosomes for 488,377 individuals. We then used the UK biobank SNPs QC file (UK biobank resource 1955) to keep only variants used for the phasing of the original UK biobank release, resulting in 670,741 variant sites.

### Validation and production datasets
We assembled two distinct datasets comprising different collections of samples of British or Irish ancestries by subsampling the original dataset with BCFtools v1.8. The first one includes all UK Biobank samples excluding the parental genomes for the $N = 1399$ validation samples for which we have both parental genomes and surrogate parents. We ran our inference on $N = 1399$ validation samples and we assessed its performance by comparing our estimates to the truth given by parental genomes. It is important to note that parental genomes have been used only at the validation stage and not during any phasing runs nor PofO inference. The second dataset includes this time all available UK Biobank samples and has been used to produce the final set of individuals with PofO inference that has been used for association testing. This includes $N = 21,484$ samples for which PofO could be inferred from surrogate parents and $N = 4909$ samples for which PofO could be directly inferred from the

trios/duos. The final dataset includes 22,652 males (85.8%) and 3741 females (14.2%).

## PofO inference step1: IBD mapping

In this first stage, we inferred PofO for alleles shared IBD with surrogate parents. To do so, we started by an initial phasing run of the data using SHAPEIT v4.2.1[15] with default parameters so that all data consists of haplotypes. Then, we designed a Hidden Markov Model (HMM)[56] to identify IBD sharing between the target haplotypes and a reference panel mixing haplotypes from two different sources: from the surrogate parents of the target (labeled as mother or father) and from unrelated samples. We aimed for such a probabilistic model for its robustness to phasing and genotyping errors compared to approaches based on exact matching such as the positional Burrows–Wheeler transform (PBWT). The model then uses a forward-backward procedure to compute, for each allele of a target haplotype, the probability of copying the allele from (i) the surrogate mother haplotypes, (ii) the surrogate father haplotypes or (iii) unrelated haplotypes. Here, we used 100 unrelated haplotypes as decoys so that the model is not forced to systematically copy from surrogate parents. When the model copies the target haplotype from a specific surrogate parent at a given locus with high probability, we can therefore infer the PofO at this locus from the parental group the surrogate parent belongs to. When the model copies from unrelated haplotypes, no inference can be made at the locus (Supplementary Figs. 1, 7 panels 1, 2). We implemented this approach in an open-source tool available on GitHub[57]. As a result of this procedure, we obtained PofO calls within haplotypes segments shared IBD with surrogate parents.

## PofO inference step2: extrapolation by phasing

In this second stage, we inferred PofO for all remaining genotyped alleles. First, we built a haplotype scaffold comprising all alleles for which we know PofO from IBD sharing with surrogate parents[14]. In other words, we forced all alleles that we knew to be co-inherited from the same ancestor to locate on the same homologous chromosome (Supplementary Figs. 1, 2B). In the scaffolds, we only included IBD tracks longer than 3 cM. We empirically determined this length on the validation set of samples by maximizing and minimizing the call rate and the error rate, respectively (see "Methods" section 'Accuracy and parameters optimization'). In addition, we considered in the haplotype scaffold only alleles having a PofO probability greater or equal to 95%. As a result, we could build paternal and maternal haplotype scaffolds that we used in a second step to rephase the entire dataset using SHAPEIT4 v4.2.1[15]. The goal of this second round of phasing is three-folds: (i) to ensure that the pool of alleles coming from the same parent land onto the same haplotype, (ii) to propagate the PofO assignment from IBD tracks to all alleles along the chromosomes and (ii) to correct long range switch errors. Point (ii) is made possible as all alleles with PofO unknown (i.e., not in IBD tracks) are phased relatively to the haplotype scaffold so that we can extrapolate their PofO from the scaffold they co-localize with (paternal/maternal). In practice, we ran SHAPEIT4 with two main options: -scaffold to specify the scaffolds of haplotypes to be used in the estimation and -bingraph to output the haplotype reconstructions together with phasing uncertainties. The latter provides the haplotype reconstructions as parsimonious graphs encapsulating phasing uncertainty so that likely haplotype pairs can be rapidly sampled without being forced to rerun the complete phasing run. As a consequence, we sampled for each target sample a 1000 haplotype pairs using different seeds and computed the probability for a given allele to be paternal or maternal from its frequency of co-localization across the 1000 pairs onto the paternal and maternal haplotype scaffolds, respectively (Supplementary Figs. 1 and 7A–H panels 3). This frequency indicates the certainty we have in phasing

and therefore is a probabilistic measurement of the confidence in the PofO assignment. For instance, a specific allele being phased with a certainty of 0.8 onto the paternal haplotype scaffold has an 80% chance to be of paternal origin. In all downstream analysis, we considered only heterozygous genotypes with a phasing probability above 0.7; a threshold that we empirically determined from the validation set of samples by maximizing and minimizing the call rate and the error rate (see "Methods" section on 'Accuracy and parameters optimization').

## PofO inference step3: extrapolation by imputation

In this third stage, we inferred PofO for untyped alleles, i.e., not included on the SNP array. To do so, we imputed the data using IMPUTE5 v1.1.4[16] with the Haplotype Reference Consortium[17] as a reference panel. As our data is phased with each haplotype assigned to a specific parent, we used the parameter -out-ap-field to run a haploid imputation of the data and separately imputed the paternal haplotype and the maternal haplotype. Of note, we filtered out all heterozygous genotypes with a phasing certainty below 0.7 prior to imputation (see previous section). As a result of haploid imputation, the PofO of imputed alleles can be probabilistically deduced from the imputation dosages: an allele imputed with a dosage of 0.85 on the paternal haplotype has 85% probability of being inherited from the father (i.e., PofO probability = 85%). Finally, we filtered out variants with an INFO score below 0.8 and obtained a dataset comprising 22,156,064 variants.

## Accuracy and parameters optimization

We used samples with both genotyped parents and groups of surrogate parents (i.e., validation set of samples N = 1399) to compute the errors in the PofO inference and to optimize the parameters of our inference method. For the trios (N = 309) and the duos (N = 1090), we determined the correct parental origin of offspring heterozygous genotypes at sites where a parent is homozygous, excluding sites with Mendel inconsistencies. We assessed the impact of two parameters on the call rate (percentage of heterozygous genotypes with PofO assignment) and the error rate (percentage of heterozygous genotypes with incorrect PofO assignment) of the PofO inference: (i) the length in centimorgan (cM) of the haplotype segments that we included in the scaffold for the second phasing run and (ii) the phasing certainty threshold we used to assume PofO to be known at heterozygous genotypes. To do so, we compute the call rate and the error rate for all combinations of the following parameters (Fig. 2a): haplotype segments of 2, 3, 5, 8, and 10 cM and threshold on the phasing certainty between 0.5 and 1.0 by steps of 0.05. Overall, we found that a phasing certainty above 0.7 and haplotype segments above 3 cM to be a good trade-off between call rate and error rate and used these values in all downstream analyses.

## Association testing for PofO

We tested 99 quantitative phenotypes of the UK biobank data set (Supplementary Data 1) from 4 phenotypic categories to allow phenome-wide association analysis of variants of interest: body size measurements, body composition by impedance, blood biochemistry and blood count. We additionally tested telomere length and birth weight which are not included in one of these categories. For telomere length, we removed individuals with reported blood cancer or malignancies. We considered only phenotypes with less than 50% of missing data. We rank-transformed each phenotype using the 'rntransform' function from the GenABEL v1.8-0R package[58]. We used the sex, age and the method used to infer the PofO of alleles as covariates (i.e., surrogate parents or direct parents). We used BOLT-LMM v2.3.4[18] to run all association tests. As recommended by the authors, we performed the model fitting only on the genotyped variants. For the additive GWAS scans, we used the -dosageFile parameter to test

imputed alleles dosages, as recommended in the documentation. For the PofO GWAS scans (i.e., maternal scan and paternal scan), we used the *-dosageFile* parameter to test the PofO dosages of alleles. In practice, we only used imputed allele dosages (i) of the paternal haplotype for the paternal-specific GWAS and (ii) of the maternal haplotype for the maternal-specific GWAS, so that PofO assignment uncertainty is propagated to association testing. We conducted a third PofO GWAS scan that compares the effect of maternally and paternally inherited minor alleles at heterozygous genotypes (i.e., differential scan). For this, we used only heterozygous genotypes with imputed minor allele dosages greater or equal to 0.95 to keep only genotypes with high confidence in the PofO. We encoded such alleles as 0 when inherited from the father and 1 when inherited from the mother. We again used the *-dosageFile* parameter to test whether the paternal and maternal alleles have differential effect at heterozygous sites with all homozygous genotypes set to missing. Prior to running association testing, we coded all variants so that we systematically tested the effects of minor alleles. We filtered out all variants with a minor allele frequency (MAF) below 1% which resulted in 7,645,537 variants for association testing.

### GWAS hits identification

We identify independent hits as having Linkage Disequilibrium (LD, $R^2$, computed with PLINK v1.90b5[55]) < 0.05 and being located at least 500 kb apart. If two hits are not independent, we select the one with the lowest p-value. We identify PofO associations as being Bonferroni significant ($p < 5 \times 10^{-08}$) in the differential scan and in the additive scan. We inferred the parent and direction of the effect using the paternal and maternal scans.

### Replication of PofO hits using the increased-variance method

We used the QUICKTEST software[11], designed to capture PofO effects as described by Hoggart et al. Software and documentation were accessed on 12.22.2021 (https://wp.unil.ch/sgg/program/quicktest/). We restricted the analysis to the subset of 443,993 genotyped samples of British or Irish ancestry. We used as covariates age, sex and the first ten PCs.

### Reporting summary

Further information on research design is available in the Nature Research Reporting Summary linked to this article.

## Data availability

The summary statistics for the four GWAS models across the 99 phenotypes are available here for download: http://poedb.dcsr.unil.ch/. The UK Biobank genetic data are available under restricted access for privacy policy reason, access can be obtained by application via the UK Biobank Access Management System (https://www.ukbiobank.ac.uk/enable-your-research/apply-for-access). Source data are provided with this paper.

## Code availability

Repository https://github.com/RJHFMSTR/PofO_inference hosts the source code of the IBD mapper used as part of this study, a full documentation of the pipeline[57], as well as the custom code used for the analysis and for the data visualization.

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

## Acknowledgements

This research has been conducted using the UK Biobank Resource under Application Number 66995 and funded by the Swiss National Science Foundation (SNSF) project grant 373 (PP00P3_176977). We thank Jonathan Marchini for highlighting the GxE interaction of the SEC16B locus and Chiara Auwerx for the discussion on the telomere length phenotype.

## Author contributions

R.J.H. and O.D. designed the study and wrote the paper. R.J.H. performed experiments. R.J.H. and O.D. developed the IBD mapping algorithm. R.J.H. and S.R. performed the imputation. R.J.H. and D.M.R. interpreted the biological relevance of the results. R.J.H., O.D., and Z.K. designed the GWAS models. This study was initiated after discussions between A.B. and O.D. The project has been supervised by O.D. All authors reviewed the final manuscript.

## Competing interests

The authors declare no competing interests.
