## [Peer Review File · Nature Communications]

Parent-of-Origin inference for biobanksREVIEWER COMMENTS

Reviewer #1 (Remarks to the Author):

Summary of paper

Parent-of-origin effects (POE) are epigenetic phenomena that confound our ability to predict phenotype from genotype. Studies accounting for POE in humans are rare/limited because most studies lack parental origin information. Few studies are designed in a way to easily determine the parental origin of alleles. There are few ways to impute/ predict the presence of POE without knowing the parental origin of alleles.

This study set out to develop and implement an approach to impute parental origins without using trios/duos or having known genealogy information. They show that under the right conditions and with sufficient sample size, their approach can impute the parental origin of alleles. Further, they demonstrate how this information can be used to detect POE in genome wide association scans. They found both known and novel candidate associations showing parent of origin effects. The logic of their approach shows how large consortia may be able to impute parental origins in future studies, thereby allowing for the potential detection of POE without the need to alter experimental design.

Summary of approach

The basic logic of this approach is that if a dataset contains a population that is sufficiently related, then for a given individual we can sort their relatives into two groups (parent 1 or parent 2). For male individuals we can use IBD on chromosome X to determine which group is maternal relatives and which group is paternal relatives. Then, we compare the individual with its maternal relatives in order to identify regions that are IBD with the maternal side. Then you phase from those regions to construct the maternal genome. Then you use the maternal genome to deduce what the paternal genome must be.

Question/Concerns/Critiques

Methods and descriptions should be changed to make it clear that the individuals for who this can be done are all male. This is stated explicitly only in the discussion. It otherwise reads as if perhaps males are used to do the grouping, but somehow the groups can be used to infer parental origins for females as well. My understanding is that only males can have their parental origins imputed by this approach.

If for a population one goes through and infers the parental origins for all the males. Couldn't one go back and use that information to try and infer parental origins for at least some of the females? As a sort of recursive imputation?

While this approach works, it only does so under specific conditions. The individuals must have enough close relatives included in the study. They showed that the kinship required is present in UK Biobank. How common is that in such data sets? Are the conditions met in the Irish populations for historical reasons? If one were to use a population with high admixture, would that render that data unusable for imputing parental origins?

On the other hand, the logic of this approach assumes that maternal relatives and paternal relatives are unrelated. I don't think it would work well in populations with sufficient inbreeding. Many mapping studies are intentionally conducted in populations with limited gene pools. Given that this approach seems to be selling itself as a means to re-analyze existing data sets, I think this limitation must be explored/and or stated prominently.

Reviewer #2 (Remarks to the Author):

This manuscript presents an elegant and efficient approach for identification of PofO effects on a range of phenotypes, without the need for parental or genealogy data. It leverages chromosome X data and IBD sharing with distant relatives to infer maternal or paternal origin of genotypes, combining haplotype inference and haploid imputation. The significance of the proposed model lies in its capacity to overcome a basic limitation for the assessment of the contribution of PofO effects to complex phenotypes, at least in men, as it allows using large datasets of unrelated individuals for examining these types of associations. As the number of biobanks worldwide continues to grow, this approach for assignment of parental source, as suggested by the authors, could be applied in multiple datasets enabling to meta-analyze findings and thus to investigate PofO associations with a wide-range of phenotypes in a scale not previously seen.

I would, however, like to share some reservations and concerns with respect to several specific aspects of this important and well-written work.

1. First and foremost, this work was carried-out in males only. The justification for this is very clear, and the hope that future availability of genetic data will allow assignment of parental status in females also is importantly acknowledged by the authors in the Discussion. However, as such, this manuscript should explicitly state at the outset that this work investigates PofO in males only (probably indicate this in its title too) and thus it can be assumed that it misses other PofO findings by not being able to assess this in females as well. Accumulating evidence for sexually dimorphic GWAS signals for anthropometric and other traits supports this assumption (e.g. <https://doi.org/10.1093/hmg/ddy327>).

2. Related to the first comment, I believe that two points require clarification. In line 549 the authors indicate they used sex, age and method of inference as covariates in association testing. However, if only males were included in the analysis why was sex used as a covariate? If the reason for this is that

among the 864 siblings that were added to the analysis (line 435) there were some females, I suggest re-considering their inclusion. I do not see a clear benefit of doing so as the number of females in the dataset is very limited. If, however, only male siblings were included this should be noted. Adding a supplemental table with the distribution of selected characteristics of the study sample, such as age, sex (?) and phenotypes of interest, could help clarify these issues.

3. The process of parental source inference is clearly described and in detail in both the Results and the Methods, and is supported by illustrations. Nevertheless, the issue of selection of targets could benefit from a bit more information. I understand that all siblings were included in the study sample, but my concern is relevant to 2nd/3rd degree relatives that could have been selected. Were targets defined as individuals with the largest number of relatives in the full dataset? Clarification is needed on how targets were actually selected (i.e. who was preferred over who) and how robust were the analyses (both the inference of parental source and the associations) to the specific selection of targets.

4. The authors reported results for PofO associations that were generated using two passes: the first included findings that were Bonferroni significant based on both the differential and additive scans and the second focused on significant associations identified the differential scan only. The maternal and paternal associations were only used to identify the direction of the effect. This yielded a rather limited number of significant putative signals ($n=14$), none of which fall in known imprinted regions. Interestingly, in the bioRxiv preprint of their work (<https://doi.org/10.1101/2021.11.03.467079>) the authors used more liberal criteria for identifying PofO signals and these yielded a much larger number of putative findings. This discrepancy raises questions on what would be the appropriate method for identifying valid signals, optimally balancing between missing important signals and avoiding false positive ones, and on the need for reciprocal replication. It is quite likely that using a highly stringent criteria where either both additive and differential associations or just differential associations are considered as significant findings, misses out associations that are essentially maternal- or paternal-specific, and not necessarily in opposite directions. The authors indeed use their approach to replicate findings of others, yet, it could have been highly helpful if they were able to use a more liberal cutoff for identifying signals and replicating these in an external dataset. I do acknowledge that the focus of this work is to present a new approach rather than identify novel PofO effects on complex traits, yet they present and discuss the associations identified and thus need to address this limitation in some way.

5. As noted above, it is quite interesting that none of the reported findings fall in imprinted regions, which are obviously the natural candidates for identifying variants with PofO effects. The authors indeed suggest that future work could focus on imprinted loci to decrease the cost of multiple testing. In fact, a previous meta-analysis focused on imprinted regions for investigating PofO effects on cardiometabolic phenotypes and identified signals for anthropometric traits. It would be interesting to see whether these findings are replicated using the proposed method.

6. Lastly, BOLT-LMM was used for analyzing PofO associations, which is a sensible choice as it accounts for relatedness. However, I wonder if PCs should nevertheless be additionally included as covariates, and if so, does it effect the reported results. If the authors believe that including PCs results in over-adjustment, it should be mentioned.

Reviewer #3 (Remarks to the Author):

The authors inferred the parent-of-origin (paternal/maternal) alleles at most GWAS SNPs for ~26,000 UK Biobank samples, and tested and identified many paternal/maternal effect genes.

The results are noteworthy since the parent-of-origin dataset is novel and not available before. Although the process is not fully automated, this strategy could be used for many other large datasets. The parent-of-origin results are interesting. Many of previous PofO effects have been confirmed here.

The methodology is sound and quite interesting (pretty novel and clever).

I only have a few modest/minor comments:

1. How does the sample size (~20,000 samples, or precisely 26,393 samples with call rate 74.5%) in this study compared to other PofO studies? The logic is if not bigger then probably not worth the big computational/analytical effort here. My guess is the sample size here is much larger than studies before and worth commenting (it is possible that the authors already mentioned but I missed reading this part)

2. What is the relatedness like among the 26,393 UK Biobank individuals? Particularly about the first-degree relatedness. It is not clear to me (I understand the algorithm started with 2nd/3rd degree relatives). Does the GWAS tool BOLT-LMM takes good care of the close relatedness (if there is any)?

3. How is the surrogate father determined? My understanding is the chr X IBD > 0.1 is used to infer the surrogate mother for a pair of males. Does chr X IBD < 0.1 indicate a surrogate father? Or a more stringent chr X IBD cutoff is actually used?

4. Can relative pairs other than a pair of males be used for inferring surrogate parents? For example, Chr X IBD > 0.99 between two females could well indicate a surrogate father. If this strategy is also used, would the sample size in this study be doubled, or at least increase by 50%? Or it can also help further increase the inference accuracy?

5. Some algorithms could be oversimplified/not well defined in description. For example, for the IBD based group inference, it is good to explicitly mention to exclude offspring, or those samples that are related to both groups.

We would like to thank all three reviewers for their useful comments and suggestions that helped us to improve and clarify the manuscript. Please find below our responses to all the concerns that have been raised. All changes in the manuscript have been highlighted using the “track changes” mode of Microsoft Word.

Reviewer #1 Remarks to the Author:

Summary of paper

Parent-of-origin effects (POE) are epigenetic phenomena that confound our ability to predict phenotype from genotype. Studies accounting for POE in humans are rare/limited because most studies lack parental origin information. Few studies are designed in a way to easily determine the parental origin of alleles. There are few ways to impute/ predict the presence of POE without knowing the parental origin of alleles.

This study set out to develop and implement an approach to impute parental origins without using trios/duos or having known genealogy information. They show that under the right conditions and with sufficient sample size, their approach can impute the parental origin of alleles. Further, they demonstrate how this information can be used to detect POE in genome wide association scans. They found both known and novel candidate associations showing parent of origin effects. The logic of their approach shows how large consortia may be able to impute parental origins in future studies, thereby allowing for the potential detection of POE without the need to alter experimental design.

Summary of approach

The basic logic of this approach is that if a dataset contains a population that is sufficiently related, then for a given individual we can sort their relatives into two groups (parent 1 or parent 2). For male individuals we can use IBD on chromosome X to determine which group is maternal relatives and which group is paternal relatives. Then, we compare the individual with its maternal relatives in order to identify regions that are IBD with the maternal side. Then you phase from those regions to construct the maternal genome. Then you use the maternal genome to deduce what the paternal genome must be.

Question/Concerns/Critiques

Q1. Methods and descriptions should be changed to make it clear that the individuals for who this can be done are all male. This is stated explicitly only in the discussion. It otherwise reads as if perhaps males are used to do the grouping, but somehow the groups can be used to infer parental origins for females as well. My understanding is that only males can have their parental origins imputed by this approach.

R1. We thank the reviewer for the comment. We have mentioned that our inference procedure is limited to male individuals in different parts of the manuscript: in the abstract (line 22), the results section (line 95), the discussion (line 361) and the method section (line 432). In addition, it is worth mentioning that we also included a substantial number of females in our dataset (N=3,741; 14.2%) by either (i) propagating parental status to females from male siblings (lines 445-450) or (ii) by directly inferring parental status from trios/duos (lines 469-470). We now highlight these numbers of males and females in the results section (line 175),

the methods section (lines 449-450 and 470-471) and we give full details in supplementary table 1.

Q2. If for a population one goes through and infers the parental origins for all the males. Couldn't one go back and use that information to try and infer parental origins for at least some of the females? As a sort of recursive imputation?

R2. We think this is a good point made by the reviewer. Indeed, to some extent, this is what we achieved when we propagated our inference results amongst siblings (lines 445-448). As long as we could assign surrogate parents to one male, we propagated the information to all his siblings, males or females, to infer the parental origin of their haplotypes.

Q3. While this approach works, it only does so under specific conditions. The individuals must have enough close relatives included in the study. They showed that the kinship required is present in UK Biobank. How common is that in such data sets? Are the conditions met in the Irish populations for historical reasons? If one were to use a population with high admixture, would that render that data unusable for imputing parental origins?

R3. The reviewer is correct. Our PofO inference depends on the availability of 2nd and 3rd degree relatives in the dataset. This is a common property of large biobanks for two main reasons. First, because the recruitment process encourages multiple members of the same family to enroll to the biobank (the procedure is often discussed within family members). Second, as biobanks grow in size, they sample a larger fraction of the population in the country and therefore involve a larger degree of relatedness between participants. For example, up to 5 millions samples from the UK will be biobanked and sequenced within the next 5 years. We should therefore expect a much larger fraction of relatedness in this dataset compared to the current UK Biobank.

We also agree with the reviewer that high admixture makes the kinship estimation more difficult. In this dataset, the software KING has been used to produce kinship estimates as it was expected to perform reasonably well given the population structure in the UK. For more admixed populations, one can use more robust estimation methods (e.g. <https://www.ncbi.nlm.nih.gov/pmc/articles/PMC4716688/>). Beyond the kinship estimation, we do not think that admixture is expected to perturb IBD mapping nor phasing.

Q4. On the other hand, the logic of this approach assumes that maternal relatives and paternal relatives are unrelated. I don't think it would work well in populations with sufficient inbreeding. Many mapping studies are intentionally conducted in populations with limited gene pools. Given that this approach seems to be selling itself as a means to re-analyze existing data sets, I think this limitation must be explored/and or stated prominently.

R4. We again agree with the reviewer, as we already mentioned in the discussion (lines 371-375): inbreeding can be a major confounding factor when making groups of surrogate parents. However, three points should be mentioned here to mitigate this potential issue. First, we think this only applies in the case of recent inbreeding as only relatives up to the third degree are used in the parental group assignments. Second, ambiguous samples could be easily spotted based on inbreeding coefficient or runs of homozygosity. Finally, we do not expect errors in the assignment to lead to false positives but instead to a decrease in statistical power when

performing association testing as shown in the results (lines 233-238, Figure 5C-D) and as commented in the discussion (lines 342-346).

Reviewer #2 (Remarks to the Author):

This manuscript presents an elegant and efficient approach for identification of PofO effects on a range of phenotypes, without the need for parental or genealogy data. It leverages chromosome X data and IBD sharing with distant relatives to infer maternal or paternal origin of genotypes, combining haplotype inference and haploid imputation. The significance of the proposed model lies in its capacity to overcome a basic limitation for the assessment of the contribution of PofO effects to complex phenotypes, at least in men, as it allows using large datasets of unrelated individuals for examining these types of associations. As the number of biobanks worldwide continues to grow, this approach for assignment of parental source, as suggested by the authors, could be applied in multiple datasets enabling to meta-analyze findings and thus to investigate PofO associations with a wide-range of phenotypes in a scale not previously seen.

I would, however, like to share some reservations and concerns with respect to several specific aspects of this important and well-written work.

Q1. First and foremost, this work was carried-out in males only. The justification for this is very clear, and the hope that future availability of genetic data will allow assignment of parental status in females also is importantly acknowledged by the authors in the Discussion. However, as such, this manuscript should explicitly state at the outset that this work investigates PofO in males only (probably indicate this in its title too) and thus it can be assumed that it misses other PofO findings by not being able to assess this in females as well. Accumulating evidence for sexually dimorphic GWAS signals for anthropometric and other traits supports this assumption (e.g. <https://doi.org/10.1093/hmg/ddy327>).

R1. In agreement with the reviewer(s), we acknowledge that there is a large imbalance between males and females, and, we now explicitly mention in the discussion that the approach we present in this study likely misses female specific PofO effects (lines 362-364).

Q2. Related to the first comment, I believe that two points require clarification. In line 549 the authors indicate they used sex, age and method of inference as covariates in association testing. However, if only males were included in the analysis why was sex used as a covariate? If the reason for this is that among the 864 siblings that were added to the analysis (line 435) there were some females, I suggest re-considering their inclusion. I do not see a clear benefit of doing so as the number of females in the dataset is very limited. If, however, only male siblings were included this should be noted. Adding a supplemental table with the distribution of selected characteristics of the study sample, such as age, sex (?) and phenotypes of interest, could help clarify these issues.

R2. We thank the reviewer for the comment. As mentioned to Reviewer 1 (see Q1/R1), we also included a substantial number of females in our dataset (N=3,741; 14.2%) by either (i) propagating parental status to females from male siblings (lines 445-448) or (ii) by directly inferring parental status from trios/duos (lines 468-470). We think this number of females to be too large to be simply removed from the dataset as they bring a ~14% increase in sample size and could be beneficial to discover non-sex specific PofO effects. We therefore used sex

as a covariate in our association tests. As suggested by the reviewer, we added a supplemental table to describe the study sample (table S1) and added the phenotypic distributions split by sex as part of the table S2.

Q3. The process of parental source inference is clearly described and in detail in both the Results and the Methods, and is supported by illustrations. Nevertheless, the issue of selection of targets could benefit from a bit more information. I understand that all siblings were included in the study sample, but my concern is relevant to 2nd/3rd degree relatives that could have been selected. Were targets defined as individuals with the largest number of relatives in the full dataset? Clarification is needed on how targets were actually selected (i.e. who was preferred over who) and how robust were the analyses (both the inference of parental source and the associations) to the specific selection of targets.

R3. We considered all individuals with at least one close relative in the dataset and did not use any filtering criterion. Assuming we have two individuals A and B related to the 3rd degree, A can be surrogate parent of B, B can be surrogate parent of A and both could be considered as targets, depending on their sex. We now mention this symmetric property in the method section (lines 420-421). Concerning the robustness of the analyses, we dedicated an entire result section on this particular point: “*Validation of the PofO inference on duos and trios*” and showed that accuracy improves with the number of relatives. We also show in Figure 5 how our associations are affected (i) by the additional samples we bring in the analysis by inference and (ii) by the percentage of errors we make in the inference.

Q4. The authors reported results for PofO associations that were generated using two passes: the first included findings that were Bonferroni significant based on both the differential and additive scans and the second focused on significant associations identified the differential scan only. The maternal and paternal associations were only used to identify the direction of the effect. This yielded a rather limited number of significant putative signals (n=14), none of which fall in known imprinted regions. Interestingly, in the bioRxiv preprint of their work (<https://doi.org/10.1101/2021.11.03.467079>) the authors used more liberal criteria for identifying PofO signals and these yielded a much larger number of putative findings. This discrepancy raises questions on what would be the appropriate method for identifying valid signals, optimally balancing between missing important signals and avoiding false positive ones, and on the need for reciprocal replication. It is quite likely that using a highly stringent criteria where either both additive and differential associations or just differential associations are considered as significant findings, misses out associations that are essentially maternal- or paternal-specific, and not necessarily in opposite directions. The authors indeed use their approach to replicate findings of others, yet, it could have been highly helpful if they were able to use a more liberal cutoff for identifying signals and replicating these in an external dataset. I do acknowledge that the focus of this work is to present a new approach rather than identify novel PofO effects on complex traits, yet they present and discuss the associations identified and thus need to address this limitation in some way.

R4. We appreciate the enthusiasm and interest of the reviewer in our work. There are multiple methods to identify PofO effects in the literature, and they usually differ at the biological and statistical levels: some focus on imprinted regions, others on additive associations, some consider the paternal and maternal scans only while others focus on the differential scan. Each

one has different implications in terms of multiple-testing; there is no consensus method for selecting PofO associations to date and additional work is definitely needed at this level.

This is the reason why, in this study, we went for a conservative solution and chose to use the most stringent approach we could think of. Despite this, we could still discover multiple new significant associations while being able to replicate most of the PofO associations discovered so far, thereby demonstrating the relevance and usefulness of our approach. To allow researchers to implement their own strategy to identify PofO associations and to replicate/annotate signals, we release all our summary statistics in a database (<https://poedb.dcsr.unil.ch/>).

We acknowledge that in a previous version of this work (on bioRxiv), we were more permissive and identified a larger number of PofO effects. Like the reviewer, we think that these additional signals need to be replicated in independent datasets. To this aim, we are now in the process of scanning other biobanks for PofO effects using the framework described in this paper.

Q5. As noted above, it is quite interesting that none of the reported findings fall in imprinted regions, which are obviously the natural candidates for identifying variants with PofO effects. The authors indeed suggest that future work could focus on imprinted loci to decrease the cost of multiple testing. In fact, a previous meta-analysis focused on imprinted regions for investigating PofO effects on cardiometabolic phenotypes and identified signals for anthropometric traits. It would be interesting to see whether these findings are replicated using the proposed method.

R5. The reviewer is right to note that all associations fall outside imprinted regions, except one (i.e. the MEG3/DLK1 locus is known to be imprinted). However, we would like to mention that (i) the knowledge of imprinting in the human genome is far from being complete and (ii) many of the signals we replicated from previous studies actually fall in imprinted regions.

Following the suggestion made by the reviewer, we now added in our replication pass the meta-analysis on cardiometabolic phenotypes mentioned by the reviewer. Out of the three main signals reported in this study, we could replicate the two paternal effects on height, located in the HLA region (same parent and direction of effects; lines 248-256; table 2C). However, we could not replicate the maternal effect on hip circumference.

Q6. Lastly, BOLT-LMM was used for analyzing PofO associations, which is a sensible choice as it accounts for relatedness. However, I wonder if PCs should nevertheless be additionally included as covariates, and if so, does it effect the reported results. If the authors believe that including PCs results in over-adjustment, it should be mentioned.

R6. It has been shown multiple times that linear mixed models or whole genome regression methods are the best approaches to account for stratification or relatedness in the data. We used this approach as this is the current consensus approach in the field (see references below), over PCA adjustment as it used to be the case a few years ago. Of note, we have been in contact with the BOLT-LMM developer to ensure that we were using BOLT-LMM optimally. In addition, we only focussed on white British samples, limiting stratification amongst samples.

References:

- <https://www.nature.com/articles/ng.3190>
- <https://www.nature.com/articles/s41588-021-00870-7>.
- <https://www.nature.com/articles/s41588-018-0144-6>

Reviewer #3 Remarks to the Author:

The authors inferred the parent-of-origin (paternal/maternal) alleles at most GWAS SNPs for ~26,000 UK Biobank samples, and tested and identified many paternal/maternal effect genes.

The results are noteworthy since the parent-of-origin dataset is novel and not available before. Although the process is not fully automated, this strategy could be used for many other large datasets. The parent-of-origin results are interesting. Many of previous PofO effects have been confirmed here.

The methodology is sound and quite interesting (pretty novel and clever).

I only have a few modest/minor comments:

Q1. How does the sample size (~20,000 samples, or precisely 26,393 samples with call rate 74.5%) in this study compared to other PofO studies? The logic is if not bigger then probably not worth the big computational/analytical effort here. My guess is the sample size here is much larger than studies before and worth commenting (it is possible that the authors already mentioned but I missed reading this part)

R1. To our knowledge, the only dataset with a larger sample size can be found in Benonisdottir et al, 2016 which includes 88,000 Icelandic samples with PofO inference made from genealogies and using height as the phenotype. Alternatively, the focus of our work was to develop a method that works on biobanks so that hundreds of phenotypes can be scanned for PofO associations, while retaining very large sample sizes. In addition, our framework can be used in other biobanks without requiring genealogy to be available (ongoing work).

Q2. What is the relatedness like among the 26,393 UK Biobank individuals? Particularly about the first-degree relatedness. It is not clear to me (I understand the algorithm started with 2nd/3rd degree relatives). Does the GWAS tool BOLT-LMM takes good care of the close relatedness (if there is any)?

R2. The overall amount of relatedness within the 26,393 individuals matches relatively well the overall amount observed in the full set of samples (see barplots below). Concerning 1st degree relatives, we have a total of 864 sibling pairs. To account for both cryptic relatedness and stratification amongst individuals, we used the whole genome regression method implemented in BOLT-LMM as it is commonly done by other researchers on the full set of samples (<https://www.nature.com/articles/s41588-018-0144-6>).

Kinship distributions between all UK Biobank samples on the left and all samples used as part of this study on the right.

Q3. How is the surrogate father determined? My understanding is the chr X IBD > 0.1 is used to infer the surrogate mother for a pair of males. Does chr X IBD < 0.1 indicate a surrogate father? Or a more stringent chr X IBD cutoff is actually used?

R3. The IBD on chromosome X is used between male targets and their relatives, regardless of their sex (both male or female relatives are considered). We clarified the text accordingly (line 433). Concerning the threshold, we only used chromosome X IBD > 0.1 to assign the maternal group. By extension, once we infer the maternal group, the parental group left is necessarily the paternal group (line 444). Values of IBD < 0.1 do not allow to distinguish parental groups, as maternal relatives can also show low IBD coefficients (Figure S6), which happens when both the target and the relative inherit different segments of the X.

Q4. Can relative pairs other than a pair of males be used for inferring surrogate parents? For example, Chr X IBD > 0.99 between two females could well indicate a surrogate father. If this strategy is also used, would the sample size in this study be doubled, or at least increase by 50%? Or it can also help further increase the inference accuracy?

R4. As mentioned above, we only consider the sex of the target sample, not the relatives. The relatives can be either male or female. The intuition of the reviewer is interesting: to also look at completely shared chromosome X copies between female target and female relative as indicative of a paternal relationship. We therefore looked at IBD sharing on chromosome X for all validation female samples (see figure below) similarly to what we did for males in the manuscript (Figure S6). Unfortunately, this shows that no clear classification could be made for female targets, unlike for male targets.

IBD sharing on chromosome X between target females and relatives regardless of their sex, stratified depending on the type of relationship between the target and the relative (paternal or maternal). This is only shown for validation samples for which parental genomes are available.

Q5. Some algorithms could be oversimplified/not well defined in description. For example, for the IBD based group inference, it is good to explicitly mention to exclude offspring, or those samples that are related to both groups.

R5. We agree with the reviewer and adapted the text accordingly (lines 404-409 and lines 419-420).

REVIEWERS' COMMENTS

Reviewer #2 (Remarks to the Author):

The authors have sufficiently addressed all concerns raised in their rebuttal letter as well as in the revised manuscript. I have not other comments.

Reviewer #3 (Remarks to the Author):

The authors have well addressed my comments.

Comments on authors' responses to Reviewer #1:

All questions are somewhat minor and mostly about technical details on how to infer paternal or maternal relatives. As Reviewer #1 well summarized (quite positively) in the beginning paragraphs (in "Summary of paper" and "Summary of approach"), the overall algorithm is sound and clear.

All questions have been well answered, especially in Q1, Q2, and mostly Q3.

As Reviewer #1 points out (and I agree), the work does have its limitations, such as the described algorithm may not work well on more complex populations (admixture/inbreeding), and the authors may have over-sold their algorithm so that it sounds like anybody can apply this algorithm to their own datasets. Although I don't think the algorithm is automated enough (cannot be published as a software tool alone), it should be sufficient for this particular dataset and analysis, and the number of phased genomes with parent-of-origin known in this project is impressive.

We would like to thank the reviewer 2 and reviewer 3 for looking again at the manuscript.

Reviewer #2 (Remarks to the Author):

The authors have sufficiently addressed all concerns raised in their rebuttal letter as well as in the revised manuscript. I have not other comments.

Reviewer #3 (Remarks to the Author):

The authors have well addressed my comments.

Comments on authors' responses to Reviewer #1:

All questions are somewhat minor and mostly about technical details on how to infer paternal or maternal relatives. As Reviewer #1 well summarized (quite positively) in the beginning paragraphs (in "Summary of paper" and "Summary of approach"), the overall algorithm is sound and clear.

All questions have been well answered, especially in Q1, Q2, and mostly Q3.

As Reviewer #1 points out (and I agree), the work does have its limitations, such as the described algorithm may not work well on more complex populations (admixture/inbreeding), and the authors may have over-sold their algorithm so that it sounds like anybody can apply this algorithm to their own datasets.

Although I don't think the algorithm is automated enough (cannot be published as a software tool alone), it should be sufficient for this particular dataset and analysis, and the number of phased genomes with parent-of-origin known in this project is impressive.

We thank the reviewer for this comment. Hereafter our edits to account for the reviewer's comments:

- *Automation*. We added some discussion mentioning the limitation of our approach (lines 392-394).
- *Inbreeding*. We mentioned the limitation of the approach concerning inbreeding in the discussion (lines 381-385).
- *Admixture*. We now mention the limitation of the approach concerning admixture in the discussion (lines 385-387).